# WHY ARE CONVOLUTIONAL NETS MORE SAMPLE-EFFICIENT THAN FULLY-CONNECTED NETS?

**Zhiyuan Li, Yi Zhang**
Princeton University
zhiyuanli,y.zhang@cs.princeton.edu

**Sanjeev Arora**
Princeton University & IAS
arora@cs.princeton.edu

## ABSTRACT

Convolutional neural networks often dominate fully-connected counterparts in generalization performance, especially on image classification tasks. This is often explained in terms of "better inductive bias." However, this has not been made mathematically rigorous, and the hurdle is that the sufficiently wide fully-connected net can always simulate the convolutional net. Thus the training algorithm plays a role. The current work describes a natural task on which a provable sample complexity gap can be shown, for standard training algorithms. We construct a single natural distribution on $\mathbb{R}^d \times \{\pm 1\}$ on which any orthogonal-invariant algorithm (i.e. fully-connected networks trained with most gradient-based methods from gaussian initialization) requires $\Omega(d^2)$ samples to generalize while $O(1)$ samples suffice for convolutional architectures. Furthermore, we demonstrate a single target function, learning which on all possible distributions leads to an $O(1)$ vs $\Omega(d^2/\varepsilon)$ gap. The proof relies on the fact that SGD on fully-connected network is orthogonal equivariant. Similar results are achieved for $\ell_2$ regression and adaptive training algorithms, e.g. Adam and AdaGrad, which are only permutation equivariant.

## 1 INTRODUCTION

Deep convolutional nets ("ConvNets") are at the center of the deep learning revolution (Krizhevsky et al., 2012; He et al., 2016; Huang et al., 2017). For many tasks, especially in vision, convolutional architectures perform significantly better their fully-connected ("FC") counterparts, at least given the same amount of training data. Practitioners explain this phenomenon at an intuitive level by pointing out that convolutional architectures have better "inductive bias", which intuitively means the following: (i) ConvNet is a better match to the underlying structure of image data, and thus are able to achieve low training loss with far fewer parameters (ii) models with *fewer* total number of parameters generalize better.

Surprisingly, the above intuition about the better inductive bias of ConvNets over FC nets has never been made mathematically rigorous. The natural way to make it rigorous would be to show explicit learning tasks that require far more training samples on FC nets than for ConvNets. (Here "task" means, as usual in learning theory, a distribution on data points, and binary labels for them generated given using a fixed labeling function.) Surprisingly, the standard repertoire of lower bound techniques in ML theory does not seem capable of demonstrating such a separation. The reason is that any ConvNet can be simulated by an FC net of sufficient width, since a training algorithm can just zero out unneeded connections and do weight sharing as needed. Thus the key issue is not an expressiveness *per se*, but the combination of *architecture plus the training algorithm*. But if the training algorithm must be accounted for, the usual hurdle arises that we lack good mathematical understanding of the dynamics of deep net training (whether FC or ConvNet). How then can one establish the limitations of "FC nets + current training algorithms"? (Indeed, many lower bound techniques in PAC learning theory are information theoretic and ignore the training algorithm.)

The current paper makes significant progress on the above problem by exhibiting simple tasks that require $\Omega(d^2)$ factor more training samples for FC nets than for ConvNets, where $d$ is the data dimension. (In fact this is shown even for 1-dimensional ConvNets; the lowerbound easily extends to 2-D ConvNets.) The lower bound holds for FC nets trained with any of the popular algorithms

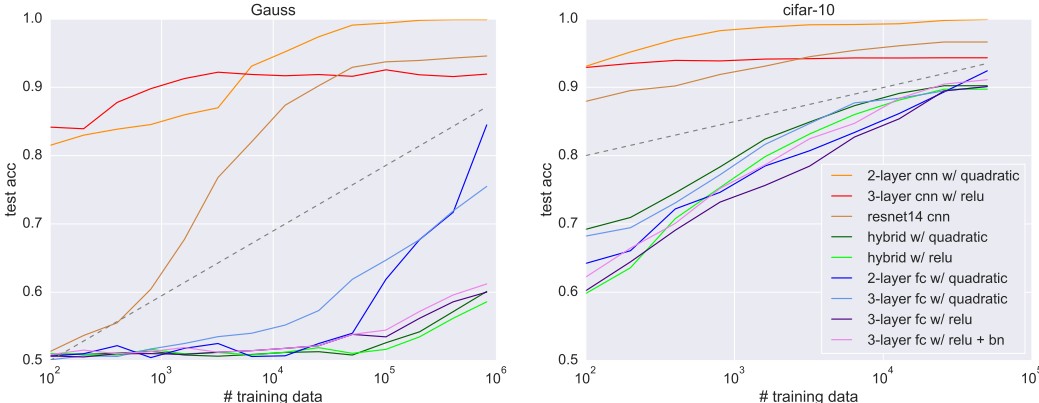

Figure 1: Comparison of generalization performance of convolutional versus fully-connected models trained by SGD. The grey dotted lines indicate separation, and we can see convolutional networks consistently outperform fully-connected networks. Here the input data are $3 \times 32 \times 32$ RGB images and the binary label indicates for each image whether the first channel has larger $\ell_2$ norm than the second one. The input images are drawn from entry-wise independent Gaussian (left) and CIFAR-10 (right). In both cases, the 3-layer convolutional networks consist of two $3 \times 3$ convolutions with 10 hidden channels, and a $3 \times 3$ convolution with a single output channel followed by global average pooling. The 3-layer fully-connected networks consist of two fully-connected layers with 10000 hidden channels and another fully-connected layer with a single output. The 2-layer versions have one less intermediate layer and have only 3072 hidden channels for each layer. The hybrid networks consist of a single fully-connected layer with 3072 channels followed by two convolutional layers with 10 channels each. **bn** stands for batch-normalization Ioffe & Szegedy (2015).

listed in Table 1. (The reader can concretely think of vanilla SGD with Gaussian initialization of network weights, though the proof allows use of momentum, $\ell_2$ regularization, and various learning rate schedules.) Our proof relies on the fact that these popular algorithms lead to an *orthogonal-equivariance* property on the trained FC nets, which says that at the end of training the FC net —no matter how deep or how wide — will make the same predictions even if we apply orthogonal transformation on all datapoints (i.e., both training and test). This notion is inspired by Ng (2004) (where it is named "orthogonal invariant"), which showed the power of logistic regression with $\ell_1$ regularization versus other learners. For a variety of learners (including kernels and FC nets) that paper described explicit tasks where the learner has $\Omega(d)$ higher sample complexity than logistic regression with $\ell_1$ regularization. The lower bound example and technique can also be extended to show a (weak) separation between FC nets and ConvNets. (See Section 4.2)

Our separation is quantitatively stronger than the result one gets using Ng (2004) because the sample complexity gap is $\Omega(d^2)$ vs $O(1)$, and not $\Omega(d)$ vs $O(1)$. But in a more subtle way our result is conceptually far stronger: the technique of Ng (2004) seems incapable of exhibiting a sample gap of more than $O(1)$ between Convnets and FC nets in our framework. The reason is that the technique of Ng (2004) can exhibit a hard task for FC nets only after *fixing* the training algorithm. But there are infinitely many training algorithms once we account for hyperparameters associated in various epochs with LR schedules, $\ell_2$ regularizer and momentum. Thus Ng (2004)'s technique cannot exclude the possibility that the hard task for "FC net + Algorithm 1" is easy for "FC net + Algorithm 2". Note that we do not claim any issues with the results claimed in Ng (2004); merely that the technique cannot lead to a proper separation between ConvNets and FC nets, when the FC nets are allowed to be trained with any of the infinitely many training algorithms. (Section 4.2 spells out in more detail the technical difference between our technique and Ng's idea.)

The reader may now be wondering what is the single task that is easy for ConvNets but hard for FC nets trained with any standard algorithm? A simple example is the following: data distribution in $\mathbb{R}^d$ is standard Gaussian, and target labeling function is the sign of $\sum_{i=1}^{d/2} x_i^2 - \sum_{i=d/2+1}^{d} x_i^2$. Figure 1 shows that this task is indeed much more difficult for FC nets. Furthermore, the task is also hard in practice for data distributions other than Gaussian; the figure shows that a sizeable performance gap exists even on CIFAR images with such a target label.

**Extension to broader class of algorithms.** The *orthogonal-equivariance* property holds for many types of practical training algorithms, but not all. Notable exceptions are adaptive gradient methods (e.g. Adam and AdaGrad), $\ell_1$ regularizer, and initialization methods that are not spherically symmetric. To prove a lower bound against FC nets with these algorithms, we identify a property, *permutation-invariance*, which is satisfied by nets trained using such algorithms. We then demonstrate a *single*

and *natural* task on $\mathbb{R}^d \times \{\pm 1\}$ that resembles real-life image texture classification, on which we prove any permutation-invariant learning algorithm requires $\Omega(d)$ training examples to generalize, while Empirical Risk Minimization with $O(1)$ examples can learn a convolutional net.

**Paper structure.** In Section 2 we discuss about related works. In section 3, we define the notation and terminologies. In Section 4, we give two warmup examples and an overview for the proof technique for the main theorem. In Section 5, we present our main results on the lower bound of orthogonal and permutation equivariant algorithms.

## 2 RELATED WORKS

Du et al. (2018) attempted to investigate the reason why convolutional nets are more sample efficient. Specifically they prove $O(1)$ samples suffice for learning a convolutional filter and also proved a $\Omega(d)$ *min-max lower bound* for learning the class of linear classifiers. Their lower bound is against learning a class of distributions, and their work fails to serve as a sample complexity separation, because their upper and lower bounds are proved on *different* classes of tasks.

Arjevani & Shamir (2016) also considered the notion of distribution-specific hardness of learning neural nets. They focused on proving running time complexity lower bounds against so-called "orthogonally invariant" and "linearly invariant" algorithms. However, here we focus on sample complexity.

Recently, there has been progress in showing lower bounds against learning with kernels. Wei et al. (2019) constructed a single task on which they proved a sample complexity separation between learning with neural networks vs. with neural tangent kernels. Notably the lower bound is specific to neural tangent kernels (Jacot et al., 2018). Relatedly, Allen-Zhu & Li (2019) showed a sample complexity lower bound against all kernels for a family of tasks, i.e., learning $k$-XOR on the hypercube.

## 3 NOTATION AND PRELIMINARIES

We will use $\mathcal{X} = \mathbb{R}^d$, $\mathcal{Y} = \{-1, 1\}$ to denote the domain of the data and label and $\mathcal{H} = \{h \mid h : \mathcal{X} \to \mathcal{Y}\}$ to denote the hypothesis class. Formally, given a joint distribution $P$, the error of a hypothesis $h \in \mathcal{H}$ is defined as $\mathrm{err}_P(h) := \mathbb{P}_{\mathbf{x}, y \sim P}[h(\mathbf{x}) \neq y]$. If $h$ is a random hypothesis, we define $\mathrm{err}_P(h) := \mathbb{P}_{\mathbf{x}, y \sim P, h}[h(\mathbf{x}) \neq y]$ for convenience. A class of joint distributions supported on $\mathcal{X} \times \mathcal{Y}$ is referred as a *problem*, $\mathcal{P}$.

We use $\|\cdot\|_2$ to denote the spectrum norm and $\|\cdot\|_F$ to denote the Frobenius norm of a matrix. We use $A \leq B$ to denote that $B - A$ is a semi-definite positive matrix. We also use $\mathcal{O}(d)$ and $\mathcal{GL}(d)$ to denote the $d$-dimensional orthogonal group and general linear group respectively. We use $B_p^{d^2}$ to denote the unit *Schatten-p* norm ball in $\mathbb{R}^{d \times d}$.

We use $N(\mu, \Sigma)$ to denote Gaussian distribution with mean $\mu$ and covariance $\Sigma$. For random variables $X$ and $Y$, we denote $X$ is equal to $Y$ in distribution by $X \stackrel{d}{=} Y$. In this work, we also always use $P_{\mathcal{X}}$ to denote the distributions on $\mathcal{X}$ and $P$ to denote the distributions supported jointly on $\mathcal{X} \times \mathcal{Y}$. Given an input distribution $P_{\mathcal{X}}$ and a hypothesis $h$, we define $P_{\mathcal{X}} \diamond h$ as the joint distribution on $\mathcal{X} \times \mathcal{Y}$, such that $(P_{\mathcal{X}} \diamond h)(S) = P(\{\mathbf{x} | (\mathbf{x}, h(\mathbf{x})) \in S\}), \forall S \subset \mathcal{X} \times \mathcal{Y}$. In other words, to sample $(X, Y) \sim P_{\mathcal{X}} \diamond h$ means to first sample $X \sim P_{\mathcal{X}}$, and then set $Y = h(X)$. For a family of input distributions $\mathcal{P}_{\mathcal{X}}$ and a hypothesis class $\mathcal{H}$, we define $\mathcal{P}_{\mathcal{X}} \diamond \mathcal{H} = \{P_{\mathcal{X}} \diamond h \mid P_{\mathcal{X}} \in \mathcal{P}_{\mathcal{X}}, h \in \mathcal{H}\}$. In this work all joint distribution $P$ can be written as $P_{\mathcal{X}} \diamond h$ for some $h$, i.e. $P_{\mathcal{Y}|\mathcal{X}}$ is deterministic.

For set $S \subset \mathcal{X}$ and 1-1 map $g : \mathcal{X} \to \mathcal{X}$, we define $g(S) = \{g(x) | x \in S\}$. We use $\circ$ to denote function composition. $(f \circ g)(x)$ is defined as $f(g(x))$, and for function classes $\mathcal{F}$, $\mathcal{G}$, $\mathcal{F} \circ \mathcal{G} = \{f \circ g \mid f \in \mathcal{F}, g \in \mathcal{G}\}$. For any distribution $P_{\mathcal{X}}$ supported on $\mathcal{X}$, we define $P_{\mathcal{X}} \circ g$ as the distribution such that $(P_{\mathcal{X}} \circ g)(S) = P_{\mathcal{X}}(g(S))$. In other words, if $X \sim P_{\mathcal{X}} \iff g^{-1}(X) \sim P_{\mathcal{X}} \circ g$, because

$$\forall S \subseteq \mathcal{X}, \quad \mathbb{P}_{X \sim P_{\mathcal{X}}}\left[g^{-1}(X) \in S\right] = \mathbb{P}_{X \sim P_{\mathcal{X}}}[X \in g(S)] = [P_{\mathcal{X}} \circ g](S).$$

---

**Algorithm 1** Iterative algorithm $\mathcal{A}$

---

**Require:** Initial parameter distribution $P_{init}$ supported in $\mathcal{W} = \mathbb{R}^m$, total iterations $T$, training
   dataset $\{\mathbf{x}_i, y_i\}_{i=1}^n$, parametric model $\mathcal{M} : \mathcal{W} \to \mathcal{H}$, iterative update rule $F(\mathbf{W}, \mathcal{M}, \{\mathbf{x}_i, y_i\}_{i=1}^n)$
**Ensure:** Hypothesis $h : \mathcal{X} \to \mathcal{Y}$.
   Sample $\mathbf{W}^{(0)} \sim P_{init}$.
   **for** $t = 0$ to $T - 1$ **do**
      $\mathbf{W}^{(t+1)} = F(\mathbf{W}^{(t)}, \mathcal{M}, \{\mathbf{x}_i, y_i\}_{i=1}^n)$.
   **return** $h = \text{sign}\left[\mathcal{M}[\mathbf{W}^{(T)}]\right]$.

---

For any joint distribution $P$ of form $P = P_{\mathcal{X}} \diamond h$, we define $P \circ g = (P_{\mathcal{X}} \circ g) \diamond (h \circ g)$. In other
words, $(X, Y) \sim P \iff (g^{-1}(X), Y) \sim P \circ g$. For any distribution class $\mathcal{P}$ and group $\mathcal{G}$ acting on
$\mathcal{X}$, we define $\mathcal{P} \circ \mathcal{G}$ as $\{P \circ g \mid P \in \mathcal{P}, g \in \mathcal{G}\}$.

**Definition 3.1.** A deterministic supervised *Learning Algorithm* $\mathcal{A}$ is a mapping from a sequence
of training data, $\{(\mathbf{x}_i, y_i)\}_{i=1}^n \in (\mathcal{X} \times \mathcal{Y})^n$, to a hypothesis $\mathcal{A}(\{(\mathbf{x}_i, y_i)\}_{i=1}^n) \in \mathcal{H} \subseteq \mathcal{Y}^{\mathcal{X}}$. The
algorithm $\mathcal{A}$ could also be randomized, in which case the output $\mathcal{A}(\{(\mathbf{x}_i, y_i)\}_{i=1}^n)$ is a distribution on
hypotheses. Two randomized algorithms $\mathcal{A}$ and $\mathcal{A}'$ are the same if for any input, their outputs have
the same distribution in function space, which is denoted by $\mathcal{A}(\{\mathbf{x}_i, y_i\}_{i=1}^n) \stackrel{d}{=} \mathcal{A}'(\{\mathbf{x}_i, y_i\}_{i=1}^n)$.

**Definition 3.2** (Equivariant Algorithms). A learning algorithm is *equivariant* under group $\mathcal{G}_{\mathcal{X}}$ (or
$\mathcal{G}_{\mathcal{X}}$-equivariant) if and only if for any dataset $\{\mathbf{x}_i, y_i\}_{i=1}^n \in (\mathcal{X} \times \mathcal{Y})^n$ and $\forall g \in \mathcal{G}_{\mathcal{X}}, \mathbf{x} \in \mathcal{X}$,
$\mathcal{A}(\{g(\mathbf{x}_i), y_i\}_{i=1}^n) \circ g = \mathcal{A}(\{\mathbf{x}_i, y_i\}_{i=1}^n)$, or $\mathcal{A}(\{g(\mathbf{x}_i), y_i\}_{i=1}^n)(g(\mathbf{x})) = [\mathcal{A}(\{\mathbf{x}_i, y_i\}_{i=1}^n)](\mathbf{x})$. [1]

**Definition 3.3** (Sample Complexity). Given a problem $\mathcal{P}$ and a randomized learning algorithm $\mathcal{A}$,
$\delta, \varepsilon \in [0, 1]$, we define the $(\varepsilon, \delta)$-*sample complexity*, denoted $\mathcal{N}(\mathcal{A}, \mathcal{P}, \varepsilon, \delta)$, as the smallest number
$n \in \mathbb{N}$ such that $\forall P \in \mathcal{P}$, w.p. $1 - \delta$ over the randomness of $\{\mathbf{x}_i, y_i\}_{i=1}^n$, $\text{err}_P(\mathcal{A}(\{\mathbf{x}_i, y_i\}_{i=1}^n)) \leq \varepsilon$.
We also define the $\varepsilon$-expected sample complexity for a problem $\mathcal{P}$, denoted $\mathcal{N}^*(\mathcal{A}, \mathcal{P}, \varepsilon)$, as the
smallest number $n \in \mathbb{N}$ such that $\forall P \in \mathcal{P}$, $\mathbb{E}_{(\mathbf{x}_i, y_i) \sim P} [\text{err}_P(\mathcal{A}(\{\mathbf{x}_i, y_i\}_{i=1}^n))] \leq \varepsilon$. By definition, we
have $\mathcal{N}^*(\mathcal{A}, \mathcal{P}, \varepsilon + \delta) \leq \mathcal{N}(\mathcal{A}, \mathcal{P}, \varepsilon, \delta) \leq \mathcal{N}^*(\mathcal{A}, \mathcal{P}, \varepsilon\delta), \; \forall \varepsilon, \delta \in [0, 1]$.

## 3.1 PARAMETRIC MODELS AND ITERATIVE ALGORITHMS

A parametric model $\mathcal{M} : \mathcal{W} \to \mathcal{H}$ is a functional mapping from weight $\mathbf{W}$ to a hypothesis
$\mathcal{M}(\cdot) : \mathcal{X} \to \mathcal{Y}$. Given a specific parametric model $\mathcal{M}$, a general iterative algorithm is defined as
Algorithm 1. In this work, we will only use the two parametric models below, FC-NN and CNN.

**FC Nets:** A $L$-layer *Fully-connected Neural Network* parameterized by its weights $\mathbf{W} = (W_1, W_2, \ldots, W_L)$ is a function $\text{FC-NN}[\cdot] : \mathbb{R}^d \to \mathbb{R}$, where $W_i \in \mathbb{R}^{d_{i-1} \times d_i}$, $d_0 = d$, and $d_L = 1$:
$$\text{FC-NN}[\mathbf{W}](\mathbf{x}) = W_L \sigma(W_{L-1} \cdots \sigma(W_2 \sigma(W_1 \mathbf{x}))).$$
Here, $\sigma : \mathbb{R} \to \mathbb{R}$ can be any function, and we abuse the notation such that $\sigma$ is also defined for vector
inputs, in the sense that $[\sigma(\mathbf{x})]_i = \sigma(x_i)$.

**ConvNets (CNN):** In this paper we will only use two layer Convolutional Neural Networks with
one channel. Suppose $d = d'r$ for some integer $d', r$, a 2-layer CNN parameterized by its weights
$\mathbf{W} = (\mathbf{w}, \mathbf{a}, b) \in \mathbb{R}^k \times \mathbb{R}^r \times \mathbb{R}$ is a function $\text{CNN}[\cdot] : \mathbb{R}^d \to \mathbb{R}$:
$$\text{CNN}[\mathbf{W}](\mathbf{x}) = \sum_{i=1}^r a_r \sigma([\mathbf{w} * \mathbf{x}]_{d'(i-1)+1:d'i}) + b,$$
where $* : \mathbb{R}^k \times \mathbb{R}^d \to \mathbb{R}^d$ is the convolution operator, defined as $[\mathbf{w} * \mathbf{x}]_i = \sum_{j=1}^k w_j x_{[i-j-1 \bmod d]+1}$,
and $\sigma : \mathbb{R}^{d'} \to \mathbb{R}$ is the composition of pooling and element-wise non-linearity.

## 3.2 EQUIVARIANCE AND TRAINING ALGORITHMS

This section gives an informal sketch of why FC nets trained with standard algorithms have certain
equivariance properties. The high level idea here is if update rule of the network, or more generally,

---

[1]For randomized algorithms, the condition becomes $\mathcal{A}(\{g(\mathbf{x}_i), y_i\}_{i=1}^n) \circ g \stackrel{d}{=} \mathcal{A}(\{\mathbf{x}_i, y_i\}_{i=1}^n)$, which is
stronger than $\mathcal{A}(\{g(\mathbf{x}_i), y_i\}_{i=1}^n)(g(\mathbf{x})) \stackrel{d}{=} [\mathcal{A}(\{\mathbf{x}_i, y_i\}_{i=1}^n)](\mathbf{x}), \forall \mathbf{x} \in \mathcal{X}$.

| Symmetry | Sign Flip | Permutation | Orthogonal | Linear |
|---|---|---|---|---|
| Matrix Group | Diagonal, $|M_{ii}| = 1$ | Permutation | Orthogonal | Invertible |
| Algorithms | AdaGrad, Adam | AdaGrad, Adam | SGD Momentum | Newton's method |
| Initialization | Symmetric distribution | i.i.d. | i.i.d. Gaussian | All zero |
| Regularization | $\ell_p$ norm | $\ell_p$ norm | $\ell_2$ norm | None |

Table 1: Examples of gradient-based equivariant training algorithms for FC networks. The initialization requirement is only for the first layer of the network.

the parametrized model, exhibits certain symmetry per step, i.e., property 2 in Theorem C.1, then by induction it will hold till the last iteration.

Taking linear regression as an example, let $\mathbf{x}_i \in \mathbb{R}^d, i \in [n]$ be the data and $\mathbf{y} \in \mathbb{R}^n$ be the labels, the GD update for $\mathcal{L}(\mathbf{w}) = \frac{1}{2} \sum_{i=1}^n (\mathbf{x}_i^\top \mathbf{w} - y_i)^2 = \frac{1}{2} \left\| \mathbf{X}^\top \mathbf{w} - \mathbf{y} \right\|_2^2$ would be $\mathbf{w}_{t+1} = F(\mathbf{w}_t, \mathbf{X}, \mathbf{y}) := \mathbf{w}_t - \eta \mathbf{X}(\mathbf{X}^\top \mathbf{w}_t - \mathbf{y})$. Now suppose there's another person trying to solve the same problem using GD with the same initial linear function, but he observes everything in a different basis, i.e., $\mathbf{X}' = U\mathbf{X}$ and $\mathbf{w}_0' = U\mathbf{w}_0$, for some orthogonal matrix $U$. Not surprisingly, he would get the same solution for GD, just in a different basis. Mathematically, this is because $\mathbf{w}_t' = U\mathbf{w}_t \implies \mathbf{w}_{t+1}' = F(\mathbf{w}_t', U\mathbf{X}, \mathbf{y}) = UF(\mathbf{w}_t, \mathbf{X}, \mathbf{y}) = U\mathbf{w}_{t+1}$. In other words, he would make the same prediction for unseen data. Thus if the initial distribution of $\mathbf{w}_0$ is the same under all basis (i.e., under rotations), e.g., gaussian $N(0, I_d)$, then $\mathbf{w}_0 \overset{d}{=} U\mathbf{w}_0 \implies F^t(\mathbf{w}_0, U\mathbf{X}, \mathbf{y}) = UF^t(\mathbf{w}_0, \mathbf{X}, \mathbf{y})$, for any iteration $t$, which means GD for linear regression is orthogonal invariant.

To show orthogonal equivariance for gradient descent on general deep FC nets, it suffices to apply the above argument on each neuron in the first layer of the FC nets. Equivariance for other training algorithms (see Table 1) can be derived in the exact same method. The rigorous statement and the proofs are deferred into Appendix C.

## 4 WARM-UP EXAMPLES AND PROOF OVERVIEW

### 4.1 EXAMPLE 1: $\Omega(d)$ LOWER BOUND AGAINST ORTHOGONAL EQUIVARIANT METHODS

We start with a simple but insightful example to how equivariance alone could suffice for some non-trivial lower bounds.

We consider a task on $\mathbb{R}^d \times \{\pm 1\}$ which is a uniform distribution on the set $\{(\mathbf{e}_i y, y) | i \in \{1, 2, \ldots, d\}, y = \pm 1\}$, denoted by $P$. Each sample from $P$ is a one-hot vector in $\mathbb{R}^d$ and the sign of the non-zero coordinate determines its label. Now imagine our goal is to learn this task using an algorithm $\mathcal{A}$. After observing a training set of $n$ labeled points $S := \{(\mathbf{x}_i, y_i)\}_{i=1}^n$, the algorithm is asked to make a prediction on an unseen test data $\mathbf{x}$, i.e., $\mathcal{A}(S)(\mathbf{x})$. Here we are concerned with *orthogonal equivariant* algorithms ——the prediction of the algorithm on the test point remains the same even if we rotate every $x_i$ and the test point $x$ by any orthogonal matrix $R$, i.e.,

$$\mathcal{A}(\{(R\mathbf{x}_i, y_i)\}_{i=1}^n)(R\mathbf{x}) \overset{d}{=} \mathcal{A}(\{(\mathbf{x}_i, y_i)\}_{i=1}^n)(\mathbf{x})$$

Now we show this algorithm fails to generalize on task $P$, if it observes only $d/2$ training examples. The main idea here is that, for a fixed training set $S$, the prediction $\mathcal{A}(\{(\mathbf{x}_i, y_i)\}_{i=1}^n)(\mathbf{x})$ is determined solely by the inner products between $\mathbf{x}$ and $\mathbf{x}_i$'s due to orthogonal equivariance, i.e., there exists a random function $f$ (which may depend on $S$) such that[2]

$$\mathcal{A}(\{(\mathbf{x}_i, y_i)\}_{i=1}^n)(\mathbf{x}) \overset{d}{=} f(\mathbf{x}^\top \mathbf{x}_1, \ldots, \mathbf{x}^\top \mathbf{x}_n)$$

But the input distribution for this task is supported on 1-hot vectors. Suppose $n < d/2$. Then at test time the probability is at least $1/2$ that the new data point $(\mathbf{x}, y) \sim P$, is such that $\mathbf{x}$ has zero inner product with all $n$ points seen in the training set $S$. This fact alone fixes the prediction of $\mathcal{A}$ to the value $f(0, \ldots, 0)$ whereas $y$ is independently and randomly chosen to be $\pm 1$. We conclude that $\mathcal{A}$ outputs the wrong answer with probability at least $1/4$.

---

[2]this can be made formal using the fact that Gram matrix determine a set of vectors up to an orthogonal transformation.

## 4.2 EXAMPLE 2: $\Omega(d^2)$ LOWER BOUND IN THE WEAK SENSE

The warm up example illustrates the main insight of (Ng, 2004), namely, that when an orthogonal equivariant algorithm is used to do learning on a certain task, it is actually being forced to simultaneously learn all orthogonal transformations of this task. Intuitively, this should make the learning much more sample-hungry compared to even Simple SGD on ConvNets, which is not orthogonal equivariant. Now we sketch why the obvious way to make this intuition precise using VC dimension (Theorem B.1) does not give a proper separation between ConvNets and FC nets, as mentioned in the Introduction.

We first fix the ground truth labeling function $h^* = \text{sign}\left[\sum_{i=1}^{d} x_i^2 - \sum_{i=d+1}^{2d} x_i^2\right]$. Algorithm $\mathcal{A}$ is orthogonal equivariant (Definition 3.2) means that for any task $P = P_\mathcal{X} \diamond h^*$, where $P_\mathcal{X}$ is the input distribution and $h^*$ is the labeling function, $\mathcal{A}$ must have the same performance on $P$ and its rotated version $P \circ U = (P_\mathcal{X} \circ U) \diamond (h^* \circ U)$, where $U$ can be any orthogonal matrix. Therefore if there's an orthogonal equivariant learning algorithm $\mathcal{A}$ that learns $h^*$ on all distributions, then $\mathcal{A}$ will also learn every the rotated copy of $h^*$, $h^* \circ U$, on every distribution $P_\mathcal{X}$, simply because $\mathcal{A}$ learns $h^*$ on distribution $P_\mathcal{X} \circ U^{-1}$. Thus $\mathcal{A}$ learns the class of labeling functions $h^* \circ \mathcal{O}(d) := \{h(\mathbf{x}) = h^*(U(\mathbf{x})) \mid U \in \mathcal{O}(d)\}$ on all distributions. (See formal statement in Theorem 5.1) By the standard lower bounds with VC dimension (See Theorem B.1), it takes at least $\Omega(\frac{\text{VCdim}(\mathcal{H} \circ \mathcal{O}(d))}{\varepsilon})$ samples for $\mathcal{A}$ to guarantee $1 - \varepsilon$ accuracy. Thus it suffices to show the VC dimension $\text{VCdim}(\mathcal{H} \circ \mathcal{O}(d)) = \Omega(d^2)$, towards a $\Omega(d^2)$ sample complexity lower bound. (Ng (2004) picks a linear thresholding function as $h^*$, and thus $\text{VCdim}(h^* \circ \mathcal{O}(d))$ is only $O(d)$.)

Formally, we have the following theorem, whose proof is deferred into Appendix D.2:

**Theorem 4.1** (All distributions, single hypothesis)**.** Let $\mathcal{P} = \{\text{all distributions}\} \diamond \{h^*\}$. For any orthogonal equivariant algorithms $\mathcal{A}$, $\mathcal{N}(\mathcal{A}, \mathcal{P}, \varepsilon, \delta) = \Omega((d^2 + \ln\frac{1}{\delta})/\varepsilon)$, while there's a 2-layer ConvNet architecture, such that $\mathcal{N}(\text{ERM}_{\text{CNN}}, \mathcal{P}, \varepsilon, \delta) = O\left(\frac{1}{\varepsilon}\left(\log\frac{1}{\varepsilon} + \log\frac{1}{\delta}\right)\right)$.

As noted in the introduction, this doesn't imply there is some task hard for every training algorithm for the FC net. The VC dimension based lower bound implies for each algorithm $\mathcal{A}$ the existence of a fixed distribution $P_\mathcal{X} \in \mathcal{P}$ and some orthogonal matrix $U_\mathcal{A}$ such that the task $(P_\mathcal{X} \circ U_\mathcal{A}^{-1}) \diamond h^*$ is hard for it. However, this does not preclude $(P_\mathcal{X} \circ U_\mathcal{A}^{-1}) \diamond h^*$ being easy for some other algorithm $\mathcal{A}'$.

## 4.3 PROOF OVERVIEW FOR FIXED DISTRIBUTION LOWER BOUNDS

At first sight, the issue highlighted above (and in the Introduction) seems difficult to get around. One possible avenue is if the hard input distribution $P_\mathcal{X}$ in the task were invariant under all orthogonal transformations, i.e., $P_\mathcal{X} = P_\mathcal{X} \circ U$ for all orthogonal matrices $U$. Unfortunately, the distribution constructed in the proof of lower bound with VC dimension is inherently discrete and cannot be made invariant to orthogonal transformations.

Our proof uses a fixed $P_\mathcal{X}$, the standard Gaussian distribution, which is indeed invariant under orthogonal transformations. The proof also uses the Benedek-Itai's lower bound, Theorem 4.2, and the main technical part of our proof is the lower bound for the the packing number $D(\mathcal{H}, \rho, \varepsilon)$ defined below (also see Equation (2)).

For function class $\mathcal{H}$, we use $\Pi_\mathcal{H}(n)$ to denote the *growth function* of $\mathcal{H}$, i.e. $\Pi_\mathcal{H}(n) := \sup_{x_1,\ldots,x_n \in \mathcal{X}} |\{(h(x_1), h(x_2), \ldots, h(x_n)) \mid h \in \mathcal{H}\}|$. Denote the VC-Dimension of $\mathcal{H}$ by $\text{VCdim}(\mathcal{H})$, by Sauer-Shelah Lemma, we know $\Pi_\mathcal{H}(n) \le \left(\frac{en}{\text{VCdim}(\mathcal{H})}\right)^{\text{VCdim}(\mathcal{H})}$ for $n \ge \text{VCdim}(\mathcal{H})$.

Let $\rho$ be a metric on $\mathcal{H}$, We define $N(\mathcal{H}, \rho, \varepsilon)$ as the $\varepsilon$-*covering number* of $\mathcal{H}$ w.r.t. $\rho$, and $D(\mathcal{H}, \rho, \varepsilon)$ as the $\varepsilon$-*packing number* of $\mathcal{H}$ w.r.t. $\rho$. For distribution $P_\mathcal{X}$, we use $\rho_\mathcal{X}(h, h') := \mathbb{P}_{X \sim P_\mathcal{X}}[h(X) \ne h'(X)]$ to denote the discrepancy between hypothesis $h$ and $h'$ w.r.t. $P_\mathcal{X}$.

**Theorem 4.2.** [Benedek-Itai's lower bound] For any algorithm $\mathcal{A}$ that $(\varepsilon, \delta)$-learns $\mathcal{H}$ with $n$ i.i.d. samples from a fixed distribution $P_\mathcal{X}$, it must hold for every

$$\Pi_\mathcal{H}(n) \ge (1 - \delta)D(\mathcal{H}, \rho_\mathcal{X}, 2\varepsilon) \tag{1}$$

Since $\Pi_\mathcal{H}(n) \le 2^n$, we have $\mathcal{N}(\mathcal{A}, P_\mathcal{X} \diamond \mathcal{H}, \varepsilon, \delta) \ge \log_2 D(\mathcal{H}, \rho_\mathcal{X}, 2\varepsilon) + \log_2(1 - \delta)$, which is the original bound from Benedek & Itai (1991). Later Long (1995) improved this bound for the regime

$n \geq \text{VCdim}(\mathcal{H})$ using Sauer-Shelah lemma, i.e.,

$$\mathcal{N}(\mathcal{A}, P_{\mathcal{X}}, \varepsilon, \delta) \geq \frac{\text{VCdim}(\mathcal{H})}{e} \left( (1-\delta)D(\mathcal{H}, \rho_{\mathcal{X}}, 2\varepsilon) \right)^{\frac{1}{\text{VCdim}(\mathcal{H})}}. \tag{2}$$

**Intuition behind Benedek-Itai's lower bound.** We first fix the data distribution as $P_{\mathcal{X}}$. Suppose the $2\varepsilon$-packing is labeled as $\{h_1, \ldots, h_{D(\mathcal{H}, \rho_{\mathcal{X}}, 2\varepsilon)}\}$ and ground truth is chosen from this $2\varepsilon$-packing, $(\varepsilon, \delta)$-learns the hypothesis $\mathcal{H}$ means the algorithm is able to recover the index of the ground truth w.p. $1 - \delta$. Thus one can think this learning process as a noisy channel which delivers $\log_2 D(\mathcal{H}, \rho_{\mathcal{X}}, 2\varepsilon)$ bits of information. Since the data distribution is fixed, unlabeled data is independent of the ground truth, and the only information source is the labels. With some information-theoretic inequalities, we can show the number of labels, or samples (i.e., bits of information) $\mathcal{N}(\mathcal{A}, P_{\mathcal{X}} \diamond \mathcal{H}, \varepsilon, \delta) \geq \log_2 D(\mathcal{H}, \rho_{\mathcal{X}}, 2\varepsilon) + \log_2(1-\delta)$. A more closer look yields Equation (2), because when $\text{VCdim}(\mathcal{H}) < \infty$, then only $\log_2 \Pi_{\mathcal{H}}(n)$ instead of $n$ bits information can be delivered.

## 5  LOWER BOUNDS

Below we first present a reduction from a special subclass of PAC learning to equivariant learning (Theorem 5.1), based on which we prove our main separation results, Theorem 4.1, 5.2, 5.3 and 5.4.

**Theorem 5.1.** If $\mathcal{P}_{\mathcal{X}}$ is a set of data distributions that is invariant under group $\mathcal{G}_{\mathcal{X}}$, i.e., $\mathcal{P}_{\mathcal{X}} \circ \mathcal{G}_{\mathcal{X}} = \mathcal{P}_{\mathcal{X}}$, then the following inequality holds. (Furthermore it becomes an equality when $\mathcal{G}_{\mathcal{X}}$ is a compact group.)

$$\inf_{\mathcal{A} \in \mathbb{A}_{\mathcal{G}_{\mathcal{X}}}} \mathcal{N}^*(\mathcal{A}, \mathcal{P}_{\mathcal{X}} \diamond \mathcal{H}, \varepsilon) \geq \inf_{\mathcal{A} \in \mathbb{A}} \mathcal{N}^*(\mathcal{A}, \mathcal{P}_{\mathcal{X}} \diamond (\mathcal{H} \circ \mathcal{G}_{\mathcal{X}}), \varepsilon) \tag{3}$$

**Remark 5.1.** The sample complexity in standard PAC learning is usually defined again hypothesis class $\mathcal{H}$ only, i.e., $\mathcal{P}_{\mathcal{X}}$ is the set of all the possible input distributions. In that case, $\mathcal{P}_{\mathcal{X}}$ is always invariant under group $\mathcal{G}_{\mathcal{X}}$, and thus Theorem 5.1 says that $\mathcal{G}_{\mathcal{X}}$-equivariant learning against hypothesis class $\mathcal{H}$ is as hard as learning against hypothesis $\mathcal{H} \circ \mathcal{G}_{\mathcal{X}}$ without equivariance constraint.

### 5.1  $\Omega(d^2)$ LOWER BOUND FOR ORTHOGONAL EQUIVARIANCE WITH A FIXED DISTRIBUTION

In this subsection we show $\Omega(d^2)$ vs $O(1)$ separation on a single task in our main theorem (Theorem 5.2). With the same proof technique, we further show we can get correct dependency on $\varepsilon$ for the lower bound, i.e., $\Omega(\frac{d^2}{\varepsilon})$, by considering a slightly larger function class, which can be learnt by ConvNets with $O(d)$ samples. We also generalize this $\Omega(d^2)$ vs $O(d)$ separation to the case of $\ell_2$ regression with a different proof technique.

**Theorem 5.2.** There's a single task, $P_X \diamond h^*$, where $h^* = \text{sign} \left[ \sum_{i=1}^d x_i^2 - \sum_{i=d+1}^{2d} x_i^2 \right]$ and $P_X = N(0, I_{2d})$ and a constant $\varepsilon_0 > 0$, independent of $d$, such that for any orthogonal equivariant algorithm $\mathcal{A}$, we have

$$\mathcal{N}^*(\mathcal{A}, P_X \diamond h^*, \varepsilon_0) = \Omega(d^2), \tag{4}$$

while there's a 2-layer ConvNet, such that $\mathcal{N}(\text{ERM}_{\text{CNN}}, P_X \diamond h^*, \varepsilon, \delta) = O\left(\frac{1}{\varepsilon} \left(\log \frac{1}{\varepsilon} + \log \frac{1}{\delta}\right)\right)$. Moreover, $\text{ERM}_{\text{CNN}}$ could be realized by gradient descent (on the second layer only).

*Proof of Theorem 5.2.* **Upper bound:** implied by upper bound in Theorem 4.1. **Lower bound:** Note that the $P_{\mathcal{X}} = N(0, I_{2d})$ is invariant under $\mathcal{O}(2d)$, by Theorem 5.1, it suffices to show that there's a constant $\varepsilon_0 > 0$ (independent of $d$), for any algorithm $\mathcal{A}$, it takes $\Omega(d^2)$ samples to learn the augmented function class $h^* \circ \mathcal{O}(2d)$ w.r.t. $P_X = N(0, I_{2d})$. Define $h_U = \text{sign} \left[ \mathbf{x}_{1:d}^\top U \mathbf{x}_{d+1:2d} \right]$, $\forall U \in \mathbb{R}^{d \times d}$, and by Lemma D.2, we have $\mathcal{H} = \{h_U \mid U \in \mathcal{O}(d)\} \subseteq h^* \circ \mathcal{O}(2d)$. Thus it suffices to a $\Omega(d^2)$ sample complexity lower bound for the sub function class $\mathcal{H}$, i.e.,

$$\mathcal{N}^*(\mathcal{A}, N(0, I_{2d}) \diamond \{\text{sign} \left[ \mathbf{x}_{1:d}^\top U \mathbf{x}_{d+1:2d} \right]\}, \varepsilon_0) = \Omega(d^2). \tag{5}$$

By Benedek&Itai's lower bound, (Benedek & Itai, 1991) (Equation (1)), we know

$$\mathcal{N}(\mathcal{A}, \mathcal{P}, \varepsilon_0, \delta) \geq \log_2 \left( (1-\delta)D(\mathcal{H}, \rho_{\mathcal{X}}, 2\varepsilon_0) \right). \tag{6}$$

By Lemma D.4, there's some constant $C$, such that $D(\mathcal{H}, \rho_{\mathcal{X}}, \varepsilon) \geq \left(\frac{C}{\varepsilon}\right)^{\frac{d(d-1)}{2}}$, $\forall \varepsilon > 0$.

The high-level idea for Lemma D.4 is to first show that $\rho_\mathcal{X}(h_U, h_V) \geq \Omega(\frac{\|U-V\|_F}{\sqrt{d}})$, and then we show the packing number of orthogonal matrices in a small neighborhood of $I_d$ w.r.t. $\frac{\|\cdot\|_F}{\sqrt{d}}$ is roughly the same as that in the tangent space of orthogonal manifold at $I_d$, i.e., the set of skew matrices, which is of dimension $\frac{d(d-1)}{2}$ and has packing number $(\frac{C}{\varepsilon})^{\frac{d(d-1)}{2}}$. The advantage of working in the tangent space is that we can apply the standard volume argument.

Setting $\delta = \frac{1}{2}$, we have $\mathcal{N}^*(\mathcal{A}, \mathcal{P}, \varepsilon_0) \geq \mathcal{N}(\mathcal{A}, \mathcal{P}, \frac{1}{2}, 2\varepsilon_0) \geq \frac{d(d-1)}{2} \log_2 \frac{C}{4\varepsilon_0} - 1 = \Omega(d^2)$. $\qquad\square$

Indeed, we can improve the above lower bound by applying Equation (2), and get

$$\mathcal{N}(\mathcal{A}, \mathcal{P}, \varepsilon, \frac{1}{2}) \geq \frac{d^2}{e}\left(\frac{1}{2}\right)^{\frac{1}{d^2}}\left(\frac{C}{\varepsilon}\right)^{\frac{1}{2}-\frac{1}{2d}} = \Omega(d^2\varepsilon^{-\frac{1}{2}+\frac{1}{2d}}). \tag{7}$$

Note that the dependency in $\varepsilon$ in Equation (7) is $\varepsilon^{-\frac{1}{2}+\frac{1}{2d}}$ is not optimal, as opposed to $\varepsilon^{-1}$ in upper bounds and other lower bounds. A possible reason for this might be that Theorem 4.2 (Long's improved version) is still not tight and it might require a tighter probabilistic upper bound for the growth number $\Pi_\mathcal{H}(n)$, at least taking $P_\mathcal{X}$ into consideration, as opposed to the current upper bound using VC dimension only. We left it as an open problem to show a single task $P$ with $\Omega(\frac{d^2}{\varepsilon})$ sample complexity for all orthogonal equivariant algorithms.

However, if the hypothesis is of VC dimension $O(d)$, using a similar idea, we can prove a $\Omega(d^2/\varepsilon)$ sample complexity lower bound for equivariant algorithms, and $O(d)$ upper bounds for ConvNets.

**Theorem 5.3** (Single distribution, multiple functions)**.** There is a problem with single input distribution, $\mathcal{P} = \{P_\mathcal{X}\} \diamond \mathcal{H} = \{N(0, I_d)\} \diamond \{\text{sign}\left[\sum_{i=1}^d \alpha_i x_i^2\right] \mid \alpha_i \in \mathbb{R}\}$, such that for any orthogonal equivariant algorithms $\mathcal{A}$ and $\varepsilon > 0$, $\mathcal{N}^*(\mathcal{A}, \mathcal{P}, \varepsilon) = \Omega(d^2/\varepsilon)$, while there's a 2-layer ConvNets architecture, such that $\mathcal{N}(\text{ERM}_{\text{CNN}}, \mathcal{P}, \varepsilon, \delta) = O(\frac{d \log \frac{1}{\varepsilon} + \log \frac{1}{\delta}}{\varepsilon})$.

Interestingly, we can show an analog of Theorem 5.3 for $\ell_2$ regression, i.e., the algorithm not only observes the signs but also the values of labels $y_i$. Here we define the $\ell_2$ loss of function $h : \mathbb{R}^d \to \mathbb{R}$ as $\ell_P(h) = \mathbb{E}_{(\mathbf{x},y)\sim P}\left[(h(\mathbf{x}) - y)^2\right]$ and the sample complexity $\mathcal{N}^*(\mathcal{A}, \mathcal{P}, \varepsilon)$ for $\ell_2$ loss similarly as the smallest number $n \in \mathbb{N}$ such that $\forall P \in \mathcal{P}$, $\mathbb{E}_{(\mathbf{x}_i,y_i)\sim P}[\ell_P(\mathcal{A}(\{\mathbf{x}_i, y_i\}_{i=1}^n))] \leq \varepsilon \mathbb{E}_{(x,y)\sim P}\left[y^2\right]$. The last term $\mathbb{E}_{(x,y)\sim P}\left[y^2\right]$ is added for normalization to avoid the scaling issue and thus any $\varepsilon > 1$ could be achieved trivially by predicting 0 for all data.

**Theorem 5.4** (Single distribution, multiple functions, $\ell_2$ regression)**.** There is a problem with single input distribution, $\mathcal{P} = \{P_\mathcal{X}\} \diamond \mathcal{H} = \{N(0, I_d)\} \diamond \{\sum_{i=1}^d \alpha_i x_i^2 \mid \alpha_i \in \mathbb{R}\}$, such that for any orthogonal equivariant algorithms $\mathcal{A}$ and $\varepsilon > 0$, $\mathcal{N}^*(\mathcal{A}, \mathcal{P}, \varepsilon) \geq \frac{d(d+3)}{2}(1 - \varepsilon) - 1$, while there's a 2-layer ConvNet architecture, such that $\mathcal{N}^*(\text{ERM}_{\text{CNN}}, \mathcal{P}, \varepsilon) \leq d$ for any $\varepsilon > 0$.

## 5.2 $\Omega(d)$ LOWER BOUND FOR PERMUTATION EQUIVARIANCE

In this subsection we will present $\Omega(d)$ lower bound for permutation equivariance via a different proof technique — direct coupling. The high-level idea of direct coupling is to show with constant probability over $(\mathbf{X}_n, \mathbf{x})$, we can find a $g \in \mathcal{G}_\mathcal{X}$, such that $g(\mathbf{X}_n) = \mathbf{X}_n$, but $\mathbf{x}$ and $g(\mathbf{x})$ has different labels, in which case no equivariant algorithm could make the correct prediction.

**Theorem 5.5.** Let $\mathbf{t}_i = \mathbf{e}_i + \mathbf{e}_{i+1}$ and $\mathbf{s}_i = \mathbf{e}_i + \mathbf{e}_{i+2}$[3] and $P$ be the uniform distribution on $\{(\mathbf{s}_i, 1)\}_{i=1}^n \cup \{(\mathbf{t}_i, -1)\}_{i=1}^n$, which is the classification problem for local textures in a 1-dimensional image with $d$ pixels. Then for any permutation equivariant algorithm $\mathcal{A}$, $\mathcal{N}(\mathcal{A}, \mathcal{P}, \frac{1}{8}, \frac{1}{8}) \geq \mathcal{N}^*(\mathcal{A}, \mathcal{P}, \frac{1}{4}) \geq \frac{d}{10}$. Meanwhile, $\mathcal{N}(\text{ERM}_{CNN}, \mathcal{P}, 0, \delta) \leq \log_2 \frac{1}{\delta} + 2$, where $\text{ERM}_{CNN}$ stands for $\text{ERM}_{CNN}$ for function class of 2-layer ConvNets.

**Remark 5.2.** The task could be understood as detecting if there are two consecutive white pixels in the black background. For proof simplicity, we take texture of length 2 as an illustrative example. It

---

[3]For vector $\mathbf{x} \in \mathbb{R}^d$, we define $x_i = x_{(i-1) \bmod d+1}$.

is straightforward to extend the same proof to more sophisticated local pattern detection problem of any constant length and to 2-dimensional images.

## 6 CONCLUSION

We rigorously justify the common intuition that ConvNets can have better inductive bias than FC nets, by constructing a single natural distribution on which any FC net requires $\Omega(d^2)$ samples to generalize if trained with most gradient-based methods starting with gaussian initialization. On the same task, $O(1)$ samples suffice for convolutional architectures. We further extend our results to permutation equivariant algorithms, including adaptive training algorithms like Adam and AdaGrad, $\ell_1$ regularization, etc. The separation becomes $\Omega(d)$ vs $O(1)$ in this case.

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

## A  SOME BASIC INEQUALITIES

**Lemma A.1.**
$$\forall x \in [-1, 1], \quad \frac{\arccos x}{\sqrt{1-x}} \geq \sqrt{2}.$$

*Proof.* Let $x = \cos(t)$, $t \in [-\pi, \pi]$, we have
$$\frac{\arccos(x)}{\sqrt{1-x}} = \frac{t}{\sqrt{1-\cos(t)}} = \frac{t}{\sqrt{2}\sin(t/2)} \geq \sqrt{2}.$$

$\square$

**Lemma A.2.** $\exists C > 0$, $\forall d \in \mathbb{N}^+$, $M \in \mathbb{R}^{d \times d}$,

$$C \|M\|_F / \sqrt{d} \leq \mathop{\mathbb{E}}_{\mathbf{x} \sim S_{d-1}} [\|M\mathbf{x}\|_2] \leq \|M\|_F / \sqrt{d}. \tag{8}$$

*Proof of Lemma A.2.*

**Upper Bound:** By Cauchy-Schwarz inequality, we have

$$\mathop{\mathbb{E}}_{\mathbf{x} \sim S_{d-1}} [\|M\mathbf{x}\|_2] \leq \sqrt{\mathop{\mathbb{E}}_{\mathbf{x} \sim S_{d-1}} \left[\|M\mathbf{x}\|_2^2\right]} = \sqrt{\text{tr}\left[M \mathop{\mathbb{E}}_{\mathbf{x} \sim S_{d-1}} [\mathbf{x}\mathbf{x}^\top] M^\top\right]} = \sqrt{\frac{\text{tr}[MM^\top]}{d}} = \frac{\|M\|_F}{\sqrt{d}}.$$

**Lower Bound:** Let $M = U\Sigma V^\top$ be the singular value decomposition of $M$, where $U, V$ are orthogonal matrices and $\Sigma$ is diagonal. Since $\|M\|_F = \|\Sigma\|_F$, and $\mathop{\mathbb{E}}_{\mathbf{x} \sim S_{d-1}} [\|M\mathbf{x}\|_2] = \mathop{\mathbb{E}}_{\mathbf{x} \sim S_{d-1}} [\|\Sigma\mathbf{x}\|_2]$, w.l.o.g., we only need to prove the lower bound for all diagonal matrices.

By Proposition 2.5.1 in (Talagrand, 2014), there's some constant $C$, such that

$$C \|\Sigma\|_F = C\sqrt{\sum_{i=1}^d \sigma_i^2} \leq \mathop{\mathbb{E}}_{\mathbf{x} \sim N(0, I_d)} \sqrt{\sum_{i=1}^d x_i^2 \sigma_i^2} = \mathop{\mathbb{E}}_{\mathbf{x} \sim N(0, I_d)} [\|M\mathbf{x}\|]_2.$$

By Cauchy-Schwarz Inequality, we have $\mathop{\mathbb{E}}_{\mathbf{x} \sim N(0, I_d)} [\|\mathbf{x}\|_2] \leq \sqrt{\mathop{\mathbb{E}}_{\mathbf{x} \sim N(0, I_d)} \left[\|\mathbf{x}\|_2^2\right]} = \sqrt{d}$. Therefore, we have

$$\begin{aligned}
&C \|\Sigma\|_F \\
&\leq \mathop{\mathbb{E}}_{\mathbf{x} \sim N(0, I_d)} [\|M\mathbf{x}\|]_2 \\
&= \mathop{\mathbb{E}}_{\hat{\mathbf{x}} \sim S_{d-1}} [\|M\hat{\mathbf{x}}\|]_2 \mathop{\mathbb{E}}_{\mathbf{x} \sim N(0, I_d)} [\|\mathbf{x}\|_2] \\
&\leq \mathop{\mathbb{E}}_{\hat{\mathbf{x}} \sim S_{d-1}} [\|M\hat{\mathbf{x}}\|]_2 \sqrt{d},
\end{aligned} \tag{9}$$

which completes the proof. $\square$

**Lemma A.1.** For any $z > 0$, we have

$$\mathop{\text{Pr}}_{x \sim N(0, \sigma)} (|x| \leq z) \leq \frac{2}{\sqrt{\pi}} \frac{z}{\sigma}$$

*Proof.*

$$\mathop{\text{Pr}}_{x \sim N(0, \sigma)} (|x| \leq z) = \int_{-z}^z \frac{1}{\sqrt{2\pi} \sigma} \exp\left(-\frac{x^2}{2\sigma^2}\right) dx \leq \sqrt{\frac{2}{\pi}} \frac{z}{\sigma}$$

$\square$

## B  UPPER AND LOWER BOUND FOR SAMPLE COMPLEXITY WITH VC DIMENSION

**Theorem B.1.** *[Blumer et al. (1989)] If learning algorithm $\mathcal{A}$ is consistent and ranged in $\mathcal{H}$, i.e. $\mathcal{A}(\{\mathbf{x}_i, y_i\}_{i=1}^n) \in \mathcal{H}$ and $\mathcal{A}(\{\mathbf{x}_i, y_i\}_{i=1}^n)(\mathbf{x}_i) = y_i, \forall i \in [n]$, then for any distribution $P_{\mathcal{X}}$ and $0 < \varepsilon, \delta < 1$, we have*

$$\mathcal{N}(\mathcal{A}, P_{\mathcal{X}} \diamond \mathcal{H}, \varepsilon, \delta) = O(\frac{VCdim(\mathcal{H}) \ln \frac{1}{\varepsilon} + \ln \frac{1}{\delta}}{\varepsilon}). \tag{10}$$

*Meanwhile, there's a distribution $P_{\mathcal{X}}$ supported on any subsets $\{x_0, \ldots, x_{d-1}\}$ which can be shattered by $\mathcal{H}$, such that for any $0 < \varepsilon, \delta < 1$ and any algorithm $\mathcal{A}$, it holds*

$$\mathcal{N}(\mathcal{A}, P_{\mathcal{X}} \diamond \mathcal{H}, \varepsilon, \delta) = \Omega(\frac{VCdim(\mathcal{H}) + \ln \frac{1}{\delta}}{\varepsilon}). \tag{11}$$

## C  EQUIVARIANCE IN ALGORITHMS

In this section, we give sufficient conditions for an iterative algorithm to be equivariant (as defined in Algorithm 1).

**Theorem C.1.** Suppose $\mathcal{G}_{\mathcal{X}}$ is a group acting on $\mathcal{X} = \mathbb{R}^d$, the iterative algorithm $\mathcal{A}$ is $\mathcal{G}_{\mathcal{X}}$-equivariant (as defined in Algorithm 1) if the following conditions are met: (proof in appendix)

1. There's a group $\mathcal{G}_{\mathcal{W}}$ acting on $\mathcal{W}$ and a group isomorphism $\tau : \mathcal{G}_{\mathcal{X}} \to \mathcal{G}_{\mathcal{W}}$, such that $\mathcal{M}[\tau(g)(\mathbf{W})](g(\mathbf{x})) = \mathcal{M}[\mathbf{W}](\mathbf{x}), \forall \mathbf{x} \in \mathcal{X}, \mathbf{W} \in \mathcal{W}, g \in \mathcal{G}$. (One can think $g$ as the rotation $U$ applied on data $\mathbf{x}$ in linear regression and $\tau(U)$ as the rotation $U$ applied on $\mathbf{w}$.)

2. Update rule $F$ is invariant under any joint group action $(g, \tau(g)), \forall g \in \mathcal{G}$. In other words, $[\tau(g)](F(\mathbf{W}, \mathcal{M}, \{\mathbf{x}_i, y_i\}_{i=1}^n)) = F([\tau(g)](\mathbf{W}), \mathcal{M}, \{g(\mathbf{x}_i), y_i\}_{i=1}^n)$.

3. The initialization $P_{init}$ is invariant under group $\mathcal{G}_{\mathcal{W}}$, i.e. $\forall g \in \mathcal{G}_{\mathcal{W}}, P_{init} = P_{init} \circ g^{-1}$.

Here we want to address that the three conditions in Theorem C.1 are natural and almost necessary. Condition 1 is the minimal expressiveness requirement for model $\mathcal{M}$ to allow equivariance. Condition 3 is required for equivariance at initialization. Condition 2 is necessary for induction.

*Proof of Theorem C.1.* $\forall g \in \mathcal{G}_{\mathcal{X}}$, we sample $\mathbf{W}^{(0)} \sim P_{init}$, and $\widetilde{\mathbf{W}}^{(0)} = \tau(g)(\mathbf{W}^{(0)})$.

By property (3), $\widetilde{\mathbf{W}}^{(0)} \stackrel{d}{=} \mathbf{W}^{(0)} \sim P_{init}$. Let $\mathbf{W}^{(t+1)} = F\left(\mathbf{W}^{(t)}, \mathcal{M}, \{\mathbf{x}_i, y_i\}_{i=1}^n\right)$ and $\widetilde{\mathbf{W}}^{(t+1)} = F\left(\widetilde{\mathbf{W}}^{(t)}, \mathcal{M}, \{g(\mathbf{x}_i), y_i\}_{i=1}^n\right)$ for $0 \le t \le T-1$, we can show $\widetilde{\mathbf{W}}^{(t)} = \tau(g)\mathbf{W}^{(t)})$ by induction using property (2). By definition of Algorithm 1, we have

$$\mathcal{A}\{\mathbf{x}_i, y_i\}_{i=1}^n \stackrel{d}{=} \mathcal{M}[\mathbf{W}^{(T)}],$$

and

$$\mathcal{M}[\widetilde{\mathbf{W}}^{(T)}] \circ g \stackrel{d}{=} \mathcal{A}(\{g(\mathbf{x}_i), y_i\}_{i=1}^n) \circ g.$$

By property (1), we have $\mathcal{M}[\widetilde{\mathbf{W}}^{(T)}](g(\mathbf{x})) = \mathcal{M}[\tau(g)(\mathbf{W}^{(T)}](g(\mathbf{x})) = \mathcal{M}[\mathbf{W}^{(T)}](\mathbf{x})$. Therefore, $\mathcal{A}(\{\mathbf{x}_i, y_i\}_{i=1}^n) \stackrel{d}{=} \mathcal{M}[\mathbf{W}^{(T)}] = \mathcal{M}[\widetilde{\mathbf{W}}^{(T)}] \circ g \stackrel{d}{=} \mathcal{A}(\{g(\mathbf{x}_i), y_i\}_{i=1}^n) \circ g$, meaning $\mathcal{A}$ is $\mathcal{G}_{\mathcal{X}}$-equivariant. $\square$

**Remark C.1.** Theorem C.1 can be extended to the stochastic case and the adaptive case which allows the algorithm to use information of the whole trajectory, i.e., the update rule could be generalized as $\mathbf{W}^{(t+1)} = F_t(\{\mathbf{W}^{(s)}\}_{s=1}^t, \mathcal{M}, \{\mathbf{x}_i, y_i\}_{i=1}^n)$, as long as (the distribution of) each $F_t$ is invariant under joint transformations.

Below are two example applications of Theorem C.1. Other results in Table 1 could be achieved in the same way.

For classification tasks, optimization algorithms often work with a differentiable surrogate loss $\ell : \mathbb{R} \to \mathbb{R}$ instead the 0-1 loss, such that $\ell(yh(\mathbf{x})) \geq \mathbb{1}[yh(\mathbf{x}) \leq 0]$, and the total loss for hypothesis $h$ and training, $\mathcal{L}(\mathcal{M}(\mathbf{W}); \{\mathbf{x}_i, y_i\}_{i=1}^n)$ is defined as $\sum_{i=1}^n \ell(y_i[\mathcal{M}(\mathbf{W})](\mathbf{x}_i))$. It's also denoted by $\mathcal{L}(\mathbf{W})$ when there's no confusion.

**Definition C.1** (Gradient Descent for FC nets). We call Algorithm 1 *Gradient Descent* if $\mathcal{M} =$ FC-NN and $F = \mathsf{GD}_\mathcal{L}$ , where $\mathsf{GD}_\mathcal{L}(\mathbf{W}) = W - \eta \nabla \mathcal{L}(\mathbf{W})$ is called the *one-step Gradient Descent update* and $\eta > 0$ is the learning rate.

---

**Algorithm 2** Gradient Descent for FC-NN (FC networks)

---

**Require:** Initial parameter distribution $P_{init}$ , total iterations $T$, training dataset $\{\mathbf{x}_i, y_i\}_{i=1}^n$, loss function $\ell$
**Ensure:** Hypothesis $h : \mathcal{X} \to \mathcal{Y}$.
  Sample $\mathbf{W}^{(0)} \sim P_{init}$.
  **for** $t = 0$ to $T - 1$ **do**
    $\mathbf{W}^{(t+1)} = \mathbf{W}^{(t)} - \eta \sum_{i=1}^n \nabla \ell(\mathsf{FC\text{-}NN}(\mathbf{W}^{(t)})(\mathbf{x}_i), y_i)$
  **return** $h = \mathrm{sign}\left[\mathsf{FC\text{-}NN}[\mathbf{W}^{(T)}]\right]$.

---

**Corollary C.2.** Fully-connected networks trained with (stochastic) gradient descent from i.i.d. Gaussian initialization is equivariant under the orthogonal group.

*Proof of Corollary C.2.* We will verify the three conditions required in Theorem C.1 one by one.

The only place we use the FC structure is for the first condition.

**Lemma C.3.** There's a subgroup $\mathcal{G}_\mathcal{W}$ of $\mathcal{O}(m)$, and a group isomorphism $\tau : \mathcal{G}_\mathcal{X} = \mathcal{O}(d) \to \mathcal{G}_\mathcal{W}$, such that $\mathsf{FC\text{-}NN}[\tau(R)(\mathbf{W})] \circ R = \mathsf{FC\text{-}NN}[\mathbf{W}]$, $\forall \mathbf{W} \in \mathcal{W}, R \in \mathcal{G}_\mathcal{X}$.

*Proof of Lemma C.3.* By definition, $\mathsf{FC\text{-}NN}[\mathbf{W}](\mathbf{x})$ could be written $\mathsf{FC\text{-}NN}[\mathbf{W}_{2:L}](\sigma(W_1\mathbf{x}))$, which implies $\mathsf{FC\text{-}NN}[\mathbf{W}](\mathbf{x}) = \mathsf{FC\text{-}NN}[W_1 R^{-1}, \mathbf{W}_{2:L}](R\mathbf{x})$, $\forall R \in \mathcal{O}(d)$, and thus we can pick $\tau(R) = O \in \mathcal{O}(m)$, where $O(\mathbf{W}) = [W_1 R^{-1}, \mathbf{W}_{2:L}]$, and $\mathcal{G}_\mathcal{W} = \tau(\mathcal{O}(d))$. $\square$

A notable property of Gradient Descent is that it is invariant under orthogonal re-parametrization. Formally, given loss function $\mathcal{L} : \mathbb{R}^m \to \mathbb{R}$ and parameters $W \in \mathbb{R}^m$, an orthogonal re-parametrization of the problem is to replace $(\mathcal{L}, W)$ by $(\mathcal{L} \circ O^{-1}, OW)$, where $O \in \mathbb{R}^{m \times m}$ is an orthogonal matrix.

**Lemma C.4** (Gradient Descent is invariant under orthogonal re-parametization). For any $\mathcal{L}, W$ and orthogonal matrix $O \in \mathbb{R}^{m \times m}$, we have $O\mathsf{GD}_\mathcal{L}(W) = \mathsf{GD}_{\mathcal{L} \circ O^{-1}}(OW)$.

*Proof of Lemma C.4.* By definition, it suffices to show that for each $i \in [n]$, and every $\mathbf{W}$ and $\mathbf{W}' = O\mathbf{W}$,

$$O\nabla_{\mathbf{W}} \ell(\mathsf{FC\text{-}NN}(\mathbf{W})(\mathbf{x}_i), y_i) = \nabla_{\mathbf{W}'} \ell(\mathsf{FC\text{-}NN}(O^{-1}\mathbf{W}')(\mathbf{x}_i), y_i),$$

which holds by chain rule. $\square$

For any $R \in \mathcal{O}(d)$, and set $O = \tau(R)$ by Lemma C.3, $[\mathcal{L} \circ O^{-1}](\mathbf{W}) = \sum_{i=1}^n \ell(y_i \mathsf{FC\text{-}NN}[O^{-1}(\mathbf{W})](\mathbf{x}_i)) = \sum_{i=1}^n \ell(y_i \mathsf{FC\text{-}NN}[\mathbf{W}](R\mathbf{x}_i))$. The second condition in Theorem C.1 is satisfied by plugging above equality into Lemma C.4.

The third condition is also satisfied since the initialization distribution is i.i.d. Gaussian, which is known to be orthogonal invariant. In fact, from the proof, it suffices to have the initialization of the first layer invariant under $\mathcal{G}_\mathcal{X}$. $\square$

**Corollary C.5.** FC nets trained with newton's method from zero initialization for the first layer and any initialization for the rest parameters is $\mathcal{GL}(d)$-equivariant, or equivariant under the group of invertible linear transformations.

Here, Netwon's method means to use $\mathsf{NT}(\mathbf{W}) = \mathbf{W} - \eta(\nabla^2 \mathcal{L}(\mathbf{W}))^{-1}\nabla \mathcal{L}(\mathbf{W})$ as the update rule and we assume $\nabla^2 \mathcal{L}(\mathbf{W})$ is invertible. Proof is deferred into Appendix, .

*Proof of Corollary C.5.* The proof is almost the same as that of Corollary C.2, except the following modifications.

**Condition 1:** If we replace the $\mathcal{O}(d), \mathcal{O}(m)$ by $\mathcal{GL}(d), \mathcal{GL}(m)$ in the statement and proof Lemma C.3, the lemma still holds.

**Condition 2:** By chain rule, one can verify the update rule Newton's method is invariant under invertible linear re-parametization, i.e. $O\mathsf{GD}_{\mathcal{L}}(W) = \mathsf{NT}_{\mathcal{L} \circ O^{-1}}(OW)$, for all invertible matrix $O$.

**Condition 3:** Since the first layer is initialized to be 0, it is invariant under any linear transformation.

$\square$

**Remark C.2.** The above results can be easily extended to the case of momentum and $L_p$ regularization. For momentum, we only need to ensure that the following update rule, $\mathbf{W}^{(t+1)} = \mathsf{GDM}(\mathbf{W}^{(t)}, \mathbf{W}^{(t-1)}, \mathcal{M}, \{\mathbf{x}_i, y_i\}_{i=1}^n) = (1+\gamma)\mathbf{W}^{(t)} - \gamma\mathbf{W}^{(t-1)} - \eta\nabla\mathcal{L}(\mathbf{W}^{(t)})$, also satisfies the property in Lemma C.4. For $L_p$ regularization, because $\|\mathbf{W}\|_p$ is independent of $\{\mathbf{x}_i, y_i\}_{i=1}^n$, we only need to ensure $\|\mathbf{W}\|_p = \|\tau(R)(\mathbf{W})\|_p$, $\forall R \in \mathcal{G}_{\mathcal{X}}$, which is easy to check when $\mathcal{G}_{\mathcal{X}}$ only contains permutation or sign-flip.

## C.1 Examples of Equivariance for non-iterative algorithms

To demonstrate the wide application of our lower bounds, we give two more examples of algorithmic equivariance where the algorithm is not iterative. The proofs are folklore.

**Definition C.2.** Given a positive semi-definite kernel $K$, the *Kernel Regression* algorithm $\mathsf{REG}_K$ is defined as:

$$\mathsf{REG}_K(\{\mathbf{x}_i, y_i\}_{i=1}^n)(\mathbf{x}) := \mathbb{1}\left[K(\mathbf{x}, \mathbf{X}_N) \cdot K(\mathbf{X}_N, \mathbf{X}_N)^\dagger \mathbf{y} \geq 0\right]$$

where $K(\mathbf{X}_N, \mathbf{X}_N) \in \mathbb{R}^{n \times n}$, $[K(\mathbf{X}_N, \mathbf{X}_N)]_{i,j} = K(\mathbf{x}_i, \mathbf{x}_j)$, $\mathbf{y} = [y_1, y_2, \ldots, y_N]^\top$ and $K(\mathbf{x}, \mathbf{X}_N) = [K(\mathbf{x}, \mathbf{x}_1), \ldots, K(\mathbf{x}, \mathbf{x}_N)]$.

**Kernel Regression:** If kernel $K$ is $\mathcal{G}_{\mathcal{X}}$-equivariant, i.e., $\forall g \in \mathcal{G}_{\mathcal{X}}, \mathbf{x}, \mathbf{y} \in \mathcal{X}, K(g(\mathbf{x}), g(\mathbf{y})) = K(\mathbf{x}, \mathbf{y})$, then algorithm $\mathsf{REG}_K$ is $\mathcal{G}_{\mathcal{X}}$-equivariant.

**ERM:** If $\mathcal{F} = \mathcal{F} \circ \mathcal{G}_{\mathcal{X}}$, and $\operatorname{argmin}_{h \in \mathcal{F}} \sum_{i=1}^n \mathbb{1}[h(\mathbf{x}_i) \neq y_i]$ is unique, then $\mathsf{ERM}_{\mathcal{F}}$ is $\mathcal{G}_{\mathcal{X}}$-equivariant.

## D Omitted proofs

### D.1 Proofs of sample complexity reduction for general equivariance

Given $\mathcal{G}_{\mathcal{X}}$-equivariant algorithm $\mathcal{A}$, by definition, $\mathcal{N}^*(\mathcal{A}, \mathcal{P}, \varepsilon) = \mathcal{N}^*(\mathcal{A}, \mathcal{P} \circ g^{-1}, \varepsilon), \forall g \in \mathcal{G}_{\mathcal{X}}$.

Consequently, we have

$$\mathcal{N}^*(\mathcal{A}, \mathcal{P}, \varepsilon) = \mathcal{N}^*(\mathcal{A}, \mathcal{P} \circ \mathcal{G}_{\mathcal{X}}, \varepsilon). \tag{12}$$

**Lemma D.1.** Let $\mathbb{A}$ be the set of all algorithms and $\mathbb{A}_{\mathcal{G}_{\mathcal{X}}}$ be the set of all $\mathcal{G}_{\mathcal{X}}$-equivariant algorithms, the following inequality holds. The equality is attained when $\mathcal{G}_{\mathcal{X}}$ is a compact group.

$$\inf_{\mathcal{A} \in \mathbb{A}_{\mathcal{G}_{\mathcal{X}}}} \mathcal{N}^*(\mathcal{A}, \mathcal{P}, \varepsilon) \geq \inf_{\mathcal{A} \in \mathbb{A}} \mathcal{N}^*(\mathcal{A}, \mathcal{P} \circ \mathcal{G}_{\mathcal{X}}, \varepsilon) \tag{13}$$

*Proof of Lemma D.1.* Take infimum over $\mathbb{A}_{\mathcal{G}_{\mathcal{X}}}$ over the both side of Equation 12, and note that $\mathbb{A}_{\mathcal{G}_{\mathcal{X}}} \subset \mathbb{A}$, Inequality 13 is immediate.

Suppose the group $\mathcal{G}_{\mathcal{X}}$ is compact and let $\mu$ be the Haar measure on it, i.e. $\forall S \subset \mathcal{G}_{\mathcal{X}}, g \in \mathcal{G}_{\mathcal{X}}, \mu(S) = \mu(g \circ S)$. We claim for each algorithm $\mathcal{A}$, the sample complexity of the following equivariant algorithm $\mathcal{A}'$ is no higher than that of $\mathcal{A}$ on $\mathcal{P} \diamond \mathcal{G}_{\mathcal{X}}$:

$$\mathcal{A}'(\{\mathbf{x}_i, y_i\}_{i=1}^n) = \mathcal{A}(\{g(\mathbf{x}_i), y_i\}_{i=1}^n) \circ g, \text{ where } g \sim \mu.$$

By the definition of Haar measure, $\mathcal{A}'$ is $\mathcal{G}_{\mathcal{X}}$-equivariant. Moreover, for any fixed $n \geq 0$, we have

$$\inf_{P \in \mathcal{P}} \mathbb{E}_{(\mathbf{x}_i, y_i) \sim P}[\operatorname{err}_P(\mathcal{A}'(\{\mathbf{x}_i, y_i\}_{i=1}^n))] = \inf_{P \in \mathcal{P}} \mathbb{E}_{g \sim \mu} \mathbb{E}_{(\mathbf{x}_i, y_i) \sim P \circ g^{-1}}[\operatorname{err}_P(\mathcal{A}(\{\mathbf{x}_i, y_i\}_{i=1}^n))]$$

$$\geq \inf_{P\in\mathcal{P}} \inf_{g\in\mathcal{G}_{\mathcal{X}}} \mathop{\mathbb{E}}_{(\mathbf{x}_i,y_i)\sim P\circ g^{-1}}\left[\mathrm{err}_P(\mathcal{A}(\{\mathbf{x}_i,y_i\}_{i=1}^n))\right] = \inf_{P\in\mathcal{P}\circ\mathcal{G}_{\mathcal{X}}} \mathop{\mathbb{E}}_{(\mathbf{x}_i,y_i)\sim P}\left[\mathrm{err}_P(\mathcal{A}(\{\mathbf{x}_i,y_i\}_{i=1}^n))\right],$$

which implies $\inf_{\mathcal{A}\in\mathbb{A}_{\mathcal{G}_{\mathcal{X}}}} \mathcal{N}^*(\mathcal{A},\mathcal{P},\varepsilon) \leq \inf_{\mathcal{A}\in\mathbb{A}} \mathcal{N}^*(\mathcal{A},\mathcal{P}\circ\mathcal{G}_{\mathcal{X}},\varepsilon)$. $\qquad\square$

*Proof of Theorem 5.1.* Simply note that $(\mathcal{P}_{\mathcal{X}}\diamond\mathcal{H})\circ\mathcal{G}_{\mathcal{X}} = \cup_{g\in\mathcal{G}_{\mathcal{X}}}(\mathcal{P}_{\mathcal{X}}\circ g)\diamond(\mathcal{H}\circ g^{-1}) = \cup_{g\in\mathcal{G}_{\mathcal{X}}}\mathcal{P}_{\mathcal{X}}\diamond(\mathcal{H}\circ g^{-1}) = \mathcal{P}_{\mathcal{X}}\diamond(\mathcal{H}\circ\mathcal{G}_{\mathcal{X}})$, the theorem is immediate from Lemma D.1. $\qquad\square$

## D.2 Proof of Theorem 4.1

**Lemma D.2.** Define $h_U = \mathrm{sign}\left[\mathbf{x}_{1:d}^\top U\, \mathbf{x}_{d+1:2d}\right], \forall U \in \mathbb{R}^{d\times d}$, we have $\mathcal{H} = \{h_U \mid U \in \mathcal{O}(d)\} \subseteq \mathrm{sign}\left[\sum_{i=1}^d x_i^2 - \sum_{i=d+1}^{2d} x_i^2\right]\circ\mathcal{O}(2d)$.

*Proof.* Note that

$$\begin{bmatrix} 0 & U \\ U^\top & 0 \end{bmatrix} = \begin{bmatrix} I_d & 0 \\ 0 & U^\top \end{bmatrix}\cdot\begin{bmatrix} 0 & I_d \\ I_d & 0 \end{bmatrix}\cdot\begin{bmatrix} I_d & 0 \\ 0 & U \end{bmatrix},$$

and

$$\begin{bmatrix} 0 & I_d \\ I_d & 0 \end{bmatrix} = \begin{bmatrix} \frac{\sqrt{2}}{2}I_d & -\frac{\sqrt{2}}{2}I_d \\ \frac{\sqrt{2}}{2}I_d & \frac{\sqrt{2}}{2}I_d \end{bmatrix}\cdot\begin{bmatrix} I_d & 0 \\ 0 & -I_d \end{bmatrix}\cdot\begin{bmatrix} \frac{\sqrt{2}}{2}I_d & \frac{\sqrt{2}}{2}I_d \\ -\frac{\sqrt{2}}{2}I_d & \frac{\sqrt{2}}{2}I_d \end{bmatrix},$$

thus for any $U\in\mathcal{O}(d), \forall\mathbf{x}\in\mathbb{R}^{2d}$,

$$\begin{aligned} h_U(\mathbf{x}) &= \mathrm{sign}\left[\mathbf{x}_{1:d}^\top U\, \mathbf{x}_{d+1:2d}\right] = \mathrm{sign}\left[\mathbf{x}^\top\begin{bmatrix} 0 & U \\ U^\top & 0 \end{bmatrix}\mathbf{x}\right] \\ &= \mathrm{sign}\left[g_U(\mathbf{x})^\top\begin{bmatrix} I_d & 0 \\ 0 & -I_d \end{bmatrix}g_U(\mathbf{x})\right] \in h^*\circ\mathcal{O}(2d), \end{aligned} \tag{14}$$

where $g_U(\mathbf{x}) = \begin{bmatrix} I_d & 0 \\ 0 & U \end{bmatrix}\cdot\begin{bmatrix} \frac{\sqrt{2}}{2}I_d & -\frac{\sqrt{2}}{2}I_d \\ \frac{\sqrt{2}}{2}I_d & \frac{\sqrt{2}}{2}I_d \end{bmatrix}\cdot\mathbf{x}$ is an orthogonal transformation on $\mathbb{R}^{2d}$.

$\qquad\square$

**Lemma D.3.** Define $h_U = \mathrm{sign}\left[\mathbf{x}_{1:d}^\top U\, \mathbf{x}_{d+1:2d}\right], \forall U\in\mathbb{R}^{d\times d}$, and $\mathcal{H} = \{h_U \mid U\in\mathcal{O}(d)\}$, we have

$$\mathrm{VCdim}(\mathcal{H}) \geq \frac{d(d-1)}{2}.$$

*Proof.* Now we claim $\mathcal{H}$ shatters $\{\mathbf{e}_i + \mathbf{e}_{d+j}\}_{1\leq i<j\leq d}$, i.e. $\mathcal{O}(d)$ can shatter $\{\mathbf{e}_i\mathbf{e}_j^\top\}_{1\leq i<j\leq d}$, or for any sign pattern $\{\sigma_{ij}\}_{1\leq i<j\leq d}$, there exists $U\in\mathcal{O}(d)$, such that $\mathrm{sign}\left[\langle U, \mathbf{e}_i\mathbf{e}_j^\top\rangle\right] = \sigma_{ij}$, which implies $\mathrm{VCdim}(\mathcal{H}) \geq \frac{d(d-1)}{2}$.

Let $\mathfrak{so}(d) = \{M \mid M = -M^\top, M\in\mathbb{R}^{d\times d}\}$, we know

$$\exp(u) = I_d + u + \frac{u^2}{2} + \cdots \in \mathcal{SO}(d),\ \forall u\in\mathfrak{so}(d).$$

Thus for any sign pattern $\{\sigma_{ij}\}_{1\leq i<j\leq d}$, let $u = \sum_{1\leq i<j\leq d}\sigma_{ij}(\mathbf{e}_i\mathbf{e}_j^\top - \mathbf{e}_j\mathbf{e}_i^\top)$ and $\lambda\to 0^+$,

$$\mathrm{sign}\left[\langle\exp(\lambda u), \mathbf{e}_i\mathbf{e}_j^\top\rangle\right] = \mathrm{sign}\left[0 + \lambda\sigma_{ij} + O(\lambda^2)\right] = \mathrm{sign}\left[\sigma_{ij} + O(\lambda)\right] = \sigma_{ij}.$$

$\qquad\square$

**Theorem 4.1** (All distributions, single hypothesis). Let $\mathcal{P} = \{\text{all distributions}\}\diamond\{h^*\}$. For any orthogonal equivariant algorithms $\mathcal{A}$, $\mathcal{N}(\mathcal{A},\mathcal{P},\varepsilon,\delta) = \Omega((d^2 + \ln\frac{1}{\delta})/\varepsilon)$, while there's a 2-layer ConvNet architecture, such that $\mathcal{N}(\mathsf{ERM}_{\mathsf{CNN}},\mathcal{P},\varepsilon,\delta) = O\left(\frac{1}{\varepsilon}\left(\log\frac{1}{\varepsilon} + \log\frac{1}{\delta}\right)\right)$.

*Proof of Theorem 4.1.* **Lower bound:** Suppose $d = 2d'$ for some integer $d'$, we construct $\mathcal{P} = \mathcal{P}_{\mathcal{X}} \diamond \mathcal{H}$, where $\mathcal{P}_{\mathcal{X}}$ is the set of all possible distributions on $\mathcal{X} = \mathbb{R}^{3k}$, and $\mathcal{H} = \{\text{sign}\left[\sum_{i=1}^{d'} x_i^2 - \sum_{i=d'+1}^{2d'} x_i^2\right]\}$. By Lemma D.2, $\mathcal{H}' = \{\text{sign}\left[\mathbf{x}_{1:d}^\top U \mathbf{x}_{d+1:2d}\right] \mid U \in \mathcal{O}(d')\} \subseteq \mathcal{H} \circ \mathcal{O}(d)$. By Theorem 5.1, we have

$$\inf_{\mathcal{A} \in \mathbb{A}_{\mathcal{G}_{\mathcal{X}}}} \mathcal{N}^*(\mathcal{A}, \mathcal{P}_{\mathcal{X}} \diamond \mathcal{H}, \varepsilon) \geq \inf_{\mathcal{A} \in \mathbb{A}} \mathcal{N}^*(\mathcal{A}, \mathcal{P}_{\mathcal{X}} \diamond (\mathcal{H} \circ \mathcal{G}_{\mathcal{X}}), \varepsilon) \geq \inf_{\mathcal{A} \in \mathbb{A}} \mathcal{N}^*(\mathcal{A}, \mathcal{P}_{\mathcal{X}} \diamond \mathcal{H}', \varepsilon) \quad (15)$$

By the lower bound in Theorem B.1, we have $\inf_{\mathcal{A} \in \mathbb{A}} \mathcal{N}^*(\mathcal{A}, \mathcal{P}_{\mathcal{X}} \diamond \mathcal{H}', \varepsilon) \geq \frac{\text{VCdim}(\mathcal{H}') + \ln \frac{1}{\delta}}{\varepsilon}$. By Lemma D.3 $\text{VCdim}(\mathcal{H}') \geq \frac{d'(d'-1)}{2} = \Omega(d^2)$.

**Upper Bound:** Take CNN as defined in Section 3.1 with $d = 2d', r = 2, k = 1, \sigma : \mathbb{R}^{d'} \to \mathbb{R}, \sigma(\mathbf{x}) = \sum_{i=1}^{d'} x_i^2$ (square activation + average pooling), we have $\mathcal{F}_{\text{CNN}} = \left\{\text{sign}\left[\sum_{i=1}^{2} a_i \sum_{j=1}^{d'} x_{(i-1)d'+j}^2 w_1^2 + b\right] \mid a_1, a_2, w_1, b \in \mathbb{R}\right\}$.

Note that $\min_{h \in \mathcal{F}_{\text{CNN}}} \text{err}_P(h) = 0, \forall P \in \mathcal{P}$, and the VC dimension of $\mathcal{F}$ is 3, by Theorem B.1, we have $\forall P \in \mathcal{P}$, w.p. $1 - \delta$, $\text{err}_P(\text{ERM}_{\mathcal{F}_{\text{CNN}}}(\{\mathbf{x}_i, y_i\}_{i=1}^n)) \leq \varepsilon$, if $n = \Omega\left(\frac{1}{\varepsilon}\left(\log \frac{1}{\varepsilon} + \log \frac{1}{\delta}\right)\right)$.

**Convergence guarantee for Gradient Descent:** We initialize all the parameters by i.i.d. standard gaussian and train the second layer by gradient descent only, i.e. set the LR of $w_1$ as 0. (Note training the second layer only is still a orthogonal-equivariant algorithm for FC nets, thus it's a valid separation.)

For any convex non-increasing surrogate loss of 0-1 loss $l$ satisfying $l(0) \geq 1, \lim_{x \to \infty} l(x) = 0$ e.g. logistic loss, we define the loss of the weight $\mathbf{W}$ as ($x_{k,i}$ is the $k$th coordinate of $\mathbf{x}_i$)

$$\mathcal{L}(\mathbf{W}) = \sum_{i=1}^n l(\mathcal{F}_{\text{CNN}}[\mathbf{W}](\mathbf{x}_i)y_i) = \sum_{i=1}^n l\left(\left(\sum_{k=1}^2 a_i \sum_{j=1}^{d'} x_{(k-1)d'+j,i}^2 w_1^2 + b\right) y_i\right),$$

which is convex in $a_i$ and $b$. Note $w_1 \neq 0$ with probability 1, which means the data are separable even with fixed first layer, i.e. $\min_{\mathbf{a},b} \mathcal{L}(\mathbf{W}) = \mathcal{L}(\mathbf{W}) |_{\mathbf{a}=\mathbf{a}^*, b=0} = 0$, where $\mathbf{a}^*$ is the ground truth. Thus with sufficiently small step size, GD converges to 0 loss solution. By the definition of surrogate loss, $\mathcal{L}(\mathbf{W}) < 1$ implies for $\mathbf{x}_i, l(\mathbf{x}_i y_i) < 1$ and thus the training error is 0. $\qquad \square$

## D.3 PROOFS OF LEMMAS FOR THEOREM 5.2

**Lemma D.4.** Define $h_U = \text{sign}\left[\mathbf{x}_{1:d}^\top U \mathbf{x}_{d+1:2d}\right]$, $\mathcal{H} = \{h_U \mid U \in \mathcal{O}(d)\}$, and $\rho(U, V) := \rho_{\mathcal{X}}(h_U, h_V) = \mathbb{P}_{\mathbf{x} \sim N(0, I_{2d})}[h_U(\mathbf{x}) \neq h_V(\mathbf{x})]$. There exists a constant $C$, such that the packing number $D(\mathcal{H}, \rho_{\mathcal{X}}, \varepsilon) = D(\mathcal{O}(d), \rho, \varepsilon) \geq \left(\frac{C}{\varepsilon}\right)^{\frac{d(d-1)}{2}}$.

*Proof of Lemma D.4.* The key idea here is to first lower bound $\rho_{\mathcal{X}}(U, V)$ by $\|U - V\|_F / \sqrt{d}$ and apply volume argument in the tangent space of $I_d$ in $\mathcal{O}(d)$. We have

$$
\begin{aligned}
\rho(h_U, h_V) &= \mathop{\mathbb{P}}_{\mathbf{x} \sim N(0, I_{2d})} [h_U(\mathbf{x}) \neq h_V(\mathbf{x})] \\
&= \mathop{\mathbb{P}}_{\mathbf{x} \sim N(0, I_{2d})} \left[ \left( \mathbf{x}_{1:d}^\top U \, \mathbf{x}_{d+1:2d} \right) \left( \mathbf{x}_{1:d}^\top V \, \mathbf{x}_{d+1:2d} \right) < 0 \right] \\
&= \frac{1}{\pi} \mathop{\mathbb{E}}_{\mathbf{x}_{1:d} \sim N(0, I_d)} \left[ \arccos \left( \frac{\mathbf{x}_{1:d}^\top U V^\top \mathbf{x}_{1:d}}{\|\mathbf{x}_{1:d}\|^2} \right) \right] \\
&\geq \frac{1}{\pi} \mathop{\mathbb{E}}_{\mathbf{x}_{1:d} \sim N(0, I_d)} \left[ \sqrt{2 - 2 \frac{\mathbf{x}_{1:d}^\top U V^\top \mathbf{x}_{1:d}}{\|\mathbf{x}_{1:d}\|^2}} \right] \quad \text{(by Lemma A.1)} \\
&= \frac{1}{\pi} \mathop{\mathbb{E}}_{\mathbf{x} \sim S_{d-1}} \left[ \sqrt{2 - 2\mathbf{x}^\top U V^\top \mathbf{x}} \right] \\
&= \frac{1}{\pi} \mathop{\mathbb{E}}_{\mathbf{x} \sim S_{d-1}} \left[ \left\| (U^\top - V^\top) \mathbf{x} \right\|_F \right] \\
&\geq C_1 \|U - V\|_F / \sqrt{d} \quad \text{(by Lemma A.2)}
\end{aligned}
\tag{16}
$$

Below we show it suffices to pack in the $0.4 \, \ell_\infty$ neighborhood of $I_d$. Let $\mathfrak{so}(d)$ be the Lie algebra of $SO(d)$, i.e., $\{M \in \mathbb{R}^{d \times d} \mid M = -M^\top\}$. We also define the *matrix exponential mapping* $\exp : \mathbb{R}^{d \times d} \to \mathbb{R}^{d \times d}$, where $\exp(A) = A + \frac{A^2}{2!} + \frac{A^3}{3!} + \cdots$. It holds that $\exp(\mathfrak{so}(d)) = \mathcal{SO}(d) \subseteq \mathcal{O}(d)$. The benefit of covering in such neighborhood is that it allows us to translate the problem into the tangent space of $I_d$ by the following lemma.

**Lemma D.5** (Implication of Lemma 4 in (Szarek, 1997)). For any matrix $A, B \in \mathfrak{so}(d)$, satisfying that $\|A\|_\infty \leq \frac{\pi}{4}, \|B\|_\infty \leq \frac{\pi}{4}$, we have
$$
0.4 \|A - B\|_F \leq \|\exp(A) - \exp(B)\|_F \leq \|A - B\|_F .
\tag{17}
$$

Therefore, we have

$$
D(\mathcal{H}, \rho_{\mathcal{X}}, \varepsilon) \geq D(\mathcal{O}(d), C_1 \left\| \cdot \right\|_F / \sqrt{d}, \varepsilon) \geq D(\mathfrak{so}(d) \cap \frac{\pi}{4} B_\infty^{d^2}, C_1 \left\| \cdot \right\|_F / \sqrt{d}, 2.5\varepsilon).
\tag{18}
$$

Note that $\mathfrak{so}(d)$ is a $\frac{d(d-1)}{2}$-dimensional subspace of $\mathbb{R}^{d^2}$, by Inverse Santalo's inequality (Lemma 3, (Ma & Wu, 2015)), we have

$$
\left( \frac{\mathrm{vol}(\mathfrak{so}(d) \cap B_\infty^{d^2})}{\mathrm{vol}(\mathfrak{so}(d) \cap B_2^{d^2})} \right)^{\frac{2}{d(d-1)}} \geq C_2 \frac{\sqrt{\dim(\mathfrak{so}(d))}}{\mathop{\mathbb{E}}_{G \sim N(0, I_{d^2})} \left[ \left\| \Pi_{\mathfrak{so}(d)}(G) \right\|_\infty \right]}.
$$

where $\mathrm{vol}(\cdot)$ is the $\frac{d(d-1)}{2}$ volume defined in the space of $\mathfrak{so}(d)$ and $\Pi_{\mathfrak{so}(d)}(G) = \frac{G - G^\top}{2}$ is the projection operator onto the subspace $\mathfrak{so}(d)$. We further have

$$
\mathop{\mathbb{E}}_{G \sim N(0, I_{d^2})} \left[ \left\| \Pi_{\mathfrak{so}(d)}(G) \right\|_\infty \right] = \mathop{\mathbb{E}}_{G \sim N(0, I_{d^2})} \left[ \left\| \frac{G - G^\top}{2} \right\|_\infty \right] \leq \mathop{\mathbb{E}}_{G \sim N(0, I_{d^2})} [\|G\|_\infty] \leq C_3 \sqrt{d},
$$

where the last inequality is by Theorem 4.4.5, Vershynin (2018).

Finally, we have

$$
\begin{aligned}
&D(\mathfrak{so}(d) \cap \frac{\pi}{4} B_\infty^{d^2}, C_1 \left\| \cdot \right\|_F / \sqrt{d}, 2.5\varepsilon) \\
=&D(\mathfrak{so}(d) \cap B_\infty^{d^2}, \left\| \cdot \right\|_F, \frac{10\sqrt{d}\varepsilon}{C_1\pi}) \\
\geq&\frac{\mathrm{vol}(\mathfrak{so}(d) \cap B_\infty^{d^2})}{\mathrm{vol}(\mathfrak{so}(d) \cap B_2^{d^2})} \times \left( \frac{C_1\pi}{10\sqrt{d}\varepsilon} \right)^{\frac{d(d-1)}{2}} \\
\geq&\left( \frac{C_1 C_2 \pi \sqrt{\frac{d(d-1)}{2}}}{10d\varepsilon} \right)^{\frac{d(d-1)}{2}} \\
:=&\left( \frac{C}{\varepsilon} \right)^{\frac{d(d-1)}{2}}
\end{aligned}
\tag{19}
$$

$\square$

## D.4 PROOF OF THEOREM 5.3

**Theorem 5.3** (Single distribution, multiple functions). There is a problem with single input distribution, $\mathcal{P} = \{P_\mathcal{X}\} \diamond \mathcal{H} = \{N(0, I_d)\} \diamond \{\mathrm{sign} \left[ \sum_{i=1}^d \alpha_i x_i^2 \right] \mid \alpha_i \in \mathbb{R}\}$, such that for any orthogonal equivariant algorithms $\mathcal{A}$ and $\varepsilon > 0$, $\mathcal{N}^*(\mathcal{A}, \mathcal{P}, \varepsilon) = \Omega(d^2/\varepsilon)$, while there's a 2-layer ConvNets architecture, such that $\mathcal{N}(\mathrm{ERM}_{\mathrm{CNN}}, \mathcal{P}, \varepsilon, \delta) = O(\frac{d \log \frac{1}{\varepsilon} + \log \frac{1}{\delta}}{\varepsilon})$.

*Proof of Theorem 5.3.* **Lower bound:** Note $\mathcal{P} = \{N(0, I_d)\} \diamond \mathcal{H}$, where $\mathcal{H} = \{\mathrm{sign} \left[ \sum_{i=1}^d \alpha_i x_i^2 \right] \mid \alpha_i \in \mathbb{R}\}$. Since $N(0, I_d)$ is invariant under all orthogonal transformations, by Theorem 5.1, $\inf\limits_{\text{equivariant } \mathcal{A}} \mathcal{N}^*(\mathcal{A}, N(0, I_d) \circ \mathcal{H}, \varepsilon_0) = \inf\limits_{\mathcal{A}} \mathcal{N}^*(\mathcal{A}, N(0, I_d) \diamond (\mathcal{H} \circ \mathcal{O}(d)), \varepsilon_0)$. Furthermore, it can be show that $\mathcal{H} \circ \mathcal{O}(d) = \{\mathrm{sign} \left[ \sum_{i,j} \beta_{ij} x_i x_j \right] \mid \beta_{ij} \in \mathbb{R}\}$, the sign functions of all quadratics in $\mathbb{R}^d$. Thus it suffices to show learning quadratic functions on Gaussian distribution needs $\Omega(d^2/\varepsilon)$ samples for any algorithm (see Lemma D.6, where we assume the dimension $d$ can be divided by 4).

**Upper bound:** Take $\mathsf{CNN}$ as defined in Section 3.1 with $d = d', r = 1, k = 1, \sigma : \mathbb{R} \to \mathbb{R}, \sigma(x) = x^2$ (square activation + no pooling), we have $\mathcal{F}_{\mathsf{CNN}} = \left\{ \mathrm{sign} \left[ \sum_{i=1}^d a_i w_1^2 x_i^2 + b \right] \mid a_i, w_1, b \in \mathbb{R} \right\} = \left\{ \mathrm{sign} \left[ \sum_{i=1}^d a_i x_i^2 + b \right] \mid a_i, b \in \mathbb{R} \right\}$.

Note that $\min\limits_{h \in \mathcal{F}_{\mathsf{CNN}}} \mathrm{err}_P(h) = 0, \forall P \in \mathcal{P}$, and the VC dimension of $\mathcal{F}$ is $d + 1$, by Theorem B.1, we have $\forall P \in \mathcal{P}$, w.p. $1 - \delta$, $\mathrm{err}_P \left( \mathrm{ERM}_{\mathcal{F}_{\mathsf{CNN}}}(\{\mathbf{x}_i, y_i\}_{i=1}^n) \right) \leq \varepsilon$, if $n = \Omega \left( \frac{1}{\varepsilon} \left( d \log \frac{1}{\varepsilon} + \log \frac{1}{\delta} \right) \right)$.

**Convergence guarantee for Gradient Descent:** We initialize all the parameters by i.i.d. standard gaussian and train the second layer by gradient descent only, i.e. set the LR of $w_1$ as 0. (Note training the second layer only is still a orthogonal-equivariant algorithm for FC nets, thus it's a valid separation.)

For any convex non-increasing surrogate loss of 0-1 loss $l$ satisfying $l(0) \geq 1, \lim_{x \to \infty} l(x) = 0$ e.g. logistic loss, we define the loss of the weight $\mathbf{W}$ as ($x_{k,i}$ is the $k$th coordinate of $\mathbf{x}_i$)

$$
\mathcal{L}(\mathbf{W}) = \sum_{i=1}^n l(\mathcal{F}_{\mathsf{CNN}}[\mathbf{W}](\mathbf{x}_i) y_i) = \sum_{i=1}^n l\left( (\sum_{k=1}^d w_1^2 a_i x_{k,i}^2 + b) y_i \right),
$$

which is convex in $a_i$ and $b$. Note $w_1 \neq 0$ with probability 1, which means the data are separable even with fixed first layer, i.e. $\min_{\mathbf{a}, b} \mathcal{L}(\mathbf{W}) = \mathcal{L}(\mathbf{W}) \mid_{\mathbf{a} = \mathbf{a}^*, b = 0} = 0$, where $\mathbf{a}^*$ is the ground truth.

Thus with sufficiently small step size, GD converges to 0 loss solution. By the definition of surrogate loss, $\mathcal{L}(\mathbf{W}) < 1$ implies for $\mathbf{x}_i$, $l(\mathbf{x}_i y_i) < 1$ and thus the training error is 0.

$\square$

### D.5 Proof of Lemma D.6

**Lemma D.6.** For $A \in \mathbb{R}^{d \times d}$, we define $M_A \in \mathbb{R}^{2d \times 2d}$ as $M_A = \begin{bmatrix} A & 0 \\ 0 & I_d \end{bmatrix}$, and $h_A : \mathbb{R}^{4d} \to \{-1, 1\}$ as $h_A(\mathbf{x}) = \text{sign}\left[\mathbf{x}_{1:2d}^{\top} M_A \mathbf{x}_{2d+1:4d}\right]$. Then for $\mathcal{H} = \{h_A \mid \forall A \in \mathbb{R}^{d \times d}\} \subseteq \{\text{sign}\left[\mathbf{x}^{\top} A \mathbf{x}\right] | \forall A \in \mathbb{R}^{4d \times 4d}\}$, satisfies that it holds that for any $d$, algorithm $\mathcal{A}$ and $\varepsilon > 0$,

$$\mathcal{N}^*(\mathcal{A}, \{N(0, I_{4d})\} \diamond \mathcal{H}, \varepsilon) = \Omega(\frac{d^2}{\varepsilon}).$$

*Proof of Lemma D.6.* Below we will prove a $\Omega\left(\left(\frac{1}{\varepsilon}\right)^{d^2}\right)$ lower bound for packing number, i.e. $D(\mathcal{H}, \rho_{\mathcal{X}}, 2\varepsilon_0) = D(\mathbb{R}^{d \times d}, \rho, 2\varepsilon_0)$, where $\rho(U, V) = \rho_{\mathcal{X}}(h_U, h_V)$. Then we can apply Long's improved version Equation (2) of Benedek-Itai's lower bound and get a $\Omega(d^2/\varepsilon)$ sample complexity lower bound. The reason that we can get the correct rate of $\varepsilon$ is that the VCdim($\mathcal{H}$) is exactly equal to the exponent of the packing number. (cf. the proof of Theorem 5.2)

Similar to the proof of Theorem 5.2, the key idea here is to first lower bound $\rho(U, V)$ by $\|U - V\|_F / \sqrt{d}$ and apply volume argument. Recall for $A \in \mathbb{R}^{d \times d}$, we define $M_A \in \mathbb{R}^{2d \times 2d}$ as $M_A = \begin{bmatrix} A & 0 \\ 0 & I_d \end{bmatrix}$, and $h_A : \mathbb{R}^{4d} \to \{-1, 1\}$ as $h_A(\mathbf{x}) = \text{sign}\left[\mathbf{x}_{1:2d}^{\top} M_A \mathbf{x}_{2d+1:4d}\right]$. Then for $\mathcal{H} = \{h_A \mid \forall A \in \mathbb{R}^{d \times d}\}$. Below we will see it suffices to lower bound the packing number of a subset of $\mathbb{R}^{d \times d}$, i.e. $I_d + 0.1B_{\infty}^{d^2}$, where $B_{\infty}^{d^2}$ is the unit spectral norm ball. Clearly $\forall \mathbf{x}, \|\mathbf{x}\|_2 = 1, \forall U \in I_d + 0.1B_{\infty}^{d^2}, 0.9 \leq \|U\mathbf{x}\|_2 \leq 1.1$.

Thus $\forall U, V \in I_d + 0.1B_{\infty}^{d^2}$ we have,

$$
\begin{aligned}
\rho_{\mathcal{X}}(h_U, h_V) &= \mathop{\mathbb{P}}_{\mathbf{x} \sim N(0, I_{4d})} [h_U(\mathbf{x}) \neq h_V(\mathbf{x})] \\
&= \mathop{\mathbb{P}}_{\mathbf{x} \sim N(0, I_{4d})} \left[\left(\mathbf{x}_{1:2d}^{\top} M_U \mathbf{x}_{2d+1:4d}\right)\left(\mathbf{x}_{1:2d}^{\top} M_V \mathbf{x}_{2d+1:4d}\right) < 0\right] \\
&= \frac{1}{\pi} \mathop{\mathbb{E}}_{\mathbf{x}_{1:2d} \sim N(0, I_{2d})} \left[\arccos\left(\frac{\mathbf{x}_{1:2d}^{\top} M_U M_V^{\top} \mathbf{x}_{1:2d}}{\|M_U^{\top} \mathbf{x}_{1:2d}\|_2 \|M_V^{\top} \mathbf{x}_{1:2d}\|_2}\right)\right] \\
&\geq \frac{1}{\pi} \mathop{\mathbb{E}}_{\mathbf{x}_{1:2d} \sim N(0, I_{2d})} \left[\sqrt{2 - 2\frac{\mathbf{x}_{1:2d}^{\top} M_U M_V^{\top} \mathbf{x}_{1:2d}}{\|M_U^{\top} \mathbf{x}_{1:2d}\|_2 \|M_V^{\top} \mathbf{x}_{1:2d}\|_2}}\right] \quad \text{(by Lemma A.1)} \\
&\geq \frac{\sqrt{2}}{1.1\pi} \mathop{\mathbb{E}}_{\mathbf{x}_{1:2d} \sim N(0, I_{2d})} \left[\sqrt{\|M_U^{\top} \mathbf{x}_{1:2d}\|_2 \|M_V^{\top} \mathbf{x}_{1:2d}\|_2 - \mathbf{x}_{1:2d}^{\top} M_U M_V^{\top} \mathbf{x}_{1:2d}}\right] \\
&= \frac{1}{1.1\pi} \mathop{\mathbb{E}}_{\mathbf{x}_{1:2d} \sim N(0, I_{2d})} \left[\sqrt{\|(M_U^{\top} - M_V^{\top})\mathbf{x}_{1:2d}\|_2^2 - (\|M_U^{\top} \mathbf{x}_{1:2d}\|_2 - \|M_V^{\top} \mathbf{x}_{1:2d}\|_2)^2}\right] \\
&\geq \frac{1}{1.1\pi} \Big(\mathop{\mathbb{E}}_{\mathbf{x}_{1:2d} \sim N(0, I_{2d})} \left[\|(M_U^{\top} - M_V^{\top})\mathbf{x}_{1:2d}\|_2\right] \\
&\quad - \mathop{\mathbb{E}}_{\mathbf{x}_{1:2d} \sim N(0, I_{2d})} \left[\left|\|M_U^{\top} \mathbf{x}_{1:2d}\|_2 - \|M_V^{\top} \mathbf{x}_{1:2d}\|_2\right|\right]\Big) \\
&\geq \frac{C_0}{1.1\pi} \mathop{\mathbb{E}}_{\mathbf{x}_{1:2d} \sim N(0, I_{2d})} \left[\|(M_U^{\top} - M_V^{\top})\mathbf{x}_{1:2d}\|_2\right] \quad \text{(by Lemma D.7)} \\
&\geq C_1 \|M_U - M_V\|_F / \sqrt{d} \quad \text{(by Lemma A.2)} \\
&= C_1 \|U - V\|_F / \sqrt{d}
\end{aligned}
$$

$$(20)$$

It remains to lower bound the packing number. We have

$$
\begin{aligned}
&\mathcal{M}(0.1 B_\infty^{d^2}, C_1 \left\| \cdot \right\|_F / \sqrt{d}, \varepsilon) \\
&\geq \frac{\mathrm{vol}(B_\infty^{d^2})}{\mathrm{vol}(B_2^{d^2})} \times \left( \frac{0.1 C_1}{\sqrt{d} \varepsilon} \right)^{d^2} \\
&\geq \left( \frac{C}{\varepsilon} \right)^{d^2},
\end{aligned}
\tag{21}
$$

for some constant $C$. The proof is completed by plugging the above bound and $\mathrm{VCdim}(\mathcal{H}) = d^2$ into Equation (2).

$\square$

**Lemma D.7.** Suppose $\mathbf{x}, \mathbf{x} \sim N(0, I_d)$, then $\forall R, S \in \mathbb{R}^{d \times d}$, we have

$$
\mathop{\mathbb{E}}_{\mathbf{x}} \left[ \left\| (R - S)\mathbf{x} \right\|_2 \right] - \mathop{\mathbb{E}}_{\mathbf{x}, \mathbf{y}} \left[ \left| \sqrt{\left\| R\mathbf{x} \right\|_2^2 + \left\| \mathbf{y} \right\|_2^2} - \sqrt{\left\| S\mathbf{x} \right\|_2^2 + \left\| \mathbf{y} \right\|_2^2} \right| \right] \geq C_0 \mathop{\mathbb{E}}_{\mathbf{x}} \left[ \left\| (R - S)\mathbf{x} \right\|_2 \right], \tag{22}
$$

for some constants $C_0$ independent of $R, S$ and $d$.

*Proof of Lemma D.7.* Note that

$$
\begin{aligned}
&\left| \sqrt{\left\| R\mathbf{x} \right\|_2^2 + \left\| \mathbf{y} \right\|_2^2} - \sqrt{\left\| S\mathbf{x} \right\|_2^2 + \left\| \mathbf{y} \right\|_2^2} \right| \\
&= \left| \left\| R\mathbf{x} \right\|_2 - \left\| S\mathbf{x} \right\|_2 \right| \frac{\left\| R\mathbf{x} \right\|_2 + \left\| S\mathbf{x} \right\|_2}{\sqrt{\left\| R\mathbf{x} \right\|_2^2 + \left\| \mathbf{y} \right\|_2^2} + \sqrt{\left\| S\mathbf{x} \right\|_2^2 + \left\| \mathbf{y} \right\|_2^2}} \\
&\leq \left\| (R - S)\mathbf{x} \right\|_2 \frac{\left\| R\mathbf{x} \right\|_2 + \left\| S\mathbf{x} \right\|_2}{\sqrt{\left\| R\mathbf{x} \right\|_2^2 + \left\| \mathbf{y} \right\|_2^2} + \sqrt{\left\| S\mathbf{x} \right\|_2^2 + \left\| \mathbf{y} \right\|_2^2}}
\end{aligned}
$$

Let $F(x, d)$ be the cdf of chi-square distribution, i.e. $F(x, d) = \mathbb{P}_{\mathbf{x}} \left[ \left\| \mathbf{x} \right\|_2^2 \leq x \right]$. Let $z = \frac{x}{d}$, we have $F(zd, d) \leq (ze^{1-z})^{d/2} \leq (ze^{1-z})^{1/2}$. Thus $\mathbb{P}_{\mathbf{y}} \left[ \left\| \mathbf{y} \right\|_2^2 \leq d/2 \right] < 1$, which implies for any $\left\| \mathbf{x} \right\|_2 \leq 10\sqrt{d}$,

$$
\begin{aligned}
&\mathop{\mathbb{E}}_{\mathbf{y}} \left[ \left| \sqrt{\left\| R\mathbf{x} \right\|_2^2 + \left\| \mathbf{y} \right\|_2^2} - \sqrt{\left\| S\mathbf{x} \right\|_2^2 + \left\| \mathbf{y} \right\|_2^2} \right| \right] \\
&\leq \left\| (R - S)\mathbf{x} \right\|_2 \mathop{\mathbb{E}}_{\mathbf{y}} \left[ \frac{\left\| R\mathbf{x} \right\|_2 + \left\| S\mathbf{x} \right\|_2}{\sqrt{\left\| R\mathbf{x} \right\|_2^2 + \left\| \mathbf{y} \right\|_2^2} + \sqrt{\left\| S\mathbf{x} \right\|_2^2 + \left\| \mathbf{y} \right\|_2^2}} \right] \\
&\leq (1 - \alpha_1) \left\| (R - S)\mathbf{x} \right\|_2,
\end{aligned}
$$

for some $0 < \alpha_1$.

Therefore, we have

$$
\begin{aligned}
&\mathop{\mathbb{E}}_{\mathbf{x}} \left[ \left\| (R - S)\mathbf{x} \right\|_2 \right] - \mathop{\mathbb{E}}_{\mathbf{x}, \mathbf{y}} \left[ \left| \sqrt{\left\| R\mathbf{x} \right\|_2^2 + \left\| \mathbf{y} \right\|_2^2} - \sqrt{\left\| S\mathbf{x} \right\|_2^2 + \left\| \mathbf{y} \right\|_2^2} \right| \right] \\
&\geq \mathop{\mathbb{E}}_{\mathbf{x}} \left[ \left\| (R - S)\mathbf{x} \right\|_2 \mathbb{1} \left[ \left\| \mathbf{x} \right\| \leq 10\sqrt{d} \right] \right] \\
&\quad - \mathop{\mathbb{E}}_{\mathbf{x}, \mathbf{y}} \left[ \left| \sqrt{\left\| R\mathbf{x} \right\|_2^2 + \left\| \mathbf{y} \right\|_2^2} - \sqrt{\left\| S\mathbf{x} \right\|_2^2 + \left\| \mathbf{y} \right\|_2^2} \right| \mathbb{1} \left[ \left\| \mathbf{x} \right\|_2 \leq 10\sqrt{d} \right] \right] \\
&\geq \alpha_1 \mathop{\mathbb{E}}_{\mathbf{x}} \left[ \left\| (R - S)\mathbf{x} \right\|_2 \mathbb{1} \left[ \left\| \mathbf{x} \right\|_2 \leq 10\sqrt{d} \right] \right]
\end{aligned}
$$

$$\geq \alpha_1 \alpha_2 \mathop{\mathbb{E}}_{\mathbf{x}} \left[ \|(R-S)\mathbf{x}\|_2 \right],$$

for some constant $\alpha_2 > 0$. Here we use the other side of the tail bound of cdf of chi-square, i.e. for $z > 1, 1 - F(zd, d) < (ze^{1-z})^{d/2} < (ze^{1-z})^{1/2}$.

$\square$

### D.6  PROOFS OF THEOREM 5.4

**Lemma D.8.** Let $M \in \mathbb{R}^{d \times d}$, we have $\mathop{\mathbb{E}}_{\mathbf{x} \sim N(0,I_d)} \left[ (\mathbf{x}^\top M \mathbf{x})^2 \right] = \left\| \frac{M+M^\top}{2} \right\|_F^2 + (\text{tr}[M])^2$.

*Proof of Lemma D.8.*

$$\mathop{\mathbb{E}}_{\mathbf{x} \sim N(0,I_d)} \left[ (\mathbf{x}^\top M \mathbf{x})^2 \right]$$

$$= \mathop{\mathbb{E}}_{\mathbf{x} \sim N(0,I_d)} \left[ \sum_{i,j,i'j'} x_i x_j x_{i'} x_{j'} M_{ij} M_{i'j'} \right]$$

$$= \sum_{i \neq j} (M_{ij}^2 + M_{ij}M_{ji} + M_{ii}M_{jj}) \left( \mathop{\mathbb{E}}_{x \sim N(0,1)} \left[ x^2 \right] \right)^2 + \sum_i M_{ii}^2 \mathop{\mathbb{E}}_{x \sim N(0,1)} \left[ x^4 \right]$$

$$= \sum_{i \neq j} (M_{ij}^2 + M_{ij}M_{ji} + M_{ii}M_{jj}) + 3 \sum_i M_{ii}^2$$

$$= \left\| \frac{M+M^\top}{2} \right\|_F^2 + (\text{tr}[M])^2$$

$\square$

**Theorem 5.4** (Single distribution, multiple functions, $\ell_2$ regression). There is a problem with single input distribution, $\mathcal{P} = \{P_{\mathcal{X}}\} \diamond \mathcal{H} = \{N(0, I_d)\} \diamond \{\sum_{i=1}^d \alpha_i x_i^2 \mid \alpha_i \in \mathbb{R}\}$, such that for any orthogonal equivariant algorithms $\mathcal{A}$ and $\varepsilon > 0$, $\mathcal{N}^*(\mathcal{A}, \mathcal{P}, \varepsilon) \geq \frac{d(d+3)}{2}(1-\varepsilon) - 1$, while there's a 2-layer ConvNet architecture, such that $\mathcal{N}^*(\text{ERM}_{\text{CNN}}, \mathcal{P}, \varepsilon) \leq d$ for any $\varepsilon > 0$.

*Proof of Theorem 5.4.* **Lower bound:** Similar to the proof of Theorem 5.3, it suffices to for any algorithm $\mathcal{A}$, $\mathcal{N}^*(\mathcal{A}, \mathcal{H} \circ \mathcal{O}(d), \varepsilon) \geq \frac{d(d+3)}{2}(1-\varepsilon) - 1$. Note that $\mathcal{H} \circ \mathcal{O}(d) = \{\sum_{i,j} \beta_{ij} x_i x_j \mid \beta_{ij} \in \mathbb{R}\}$ is the set of all quadratic functions. For convenience we denote $h_M(\mathbf{x}) = \mathbf{x}^\top M \mathbf{x}, \forall M \in \mathbb{R}^{d \times d}$. Now we claim quadratic functions such that any learning algorithm $\mathcal{A}$ taking at most $n$ samples must suffer $\frac{d(d+1)}{2} - n$ loss if the ground truth quadratic function is sampled from i.i.d. gaussian. Moreover, the loss is at most $\frac{d(d+3)}{2}$ for the trivial algorithm always predicting $0$. In other words, if the expected relative error $\varepsilon \leq \frac{\frac{d(d+1)}{2} - n}{\frac{d(d+3)}{2}}$, we must have the expected sample complexity $\mathcal{N}^*(\mathcal{A}, \mathcal{P}, \varepsilon) \geq n$. That is $\mathcal{N}^*(\mathcal{A}, \mathcal{P}, \varepsilon) \geq \frac{d(d+3)}{2}(1-\varepsilon) - 1$.

(1). Upper bound for $\mathbb{E}\left[y^2\right]$. By Lemma D.8,

$$\mathop{\mathbb{E}}_{M \sim N(0,I_{d^2})} \mathop{\mathbb{E}}_{\mathbf{x} \sim P_X, y = \mathbf{x}^\top M \mathbf{x}} \left[y^2\right] = \mathop{\mathbb{E}}_{M \sim N(0,I_{d^2})} \left[ \left\| \frac{M+M^\top}{2} \right\|_F^2 + (\text{tr}[M])^2 \right] = d + d + \frac{d(d-1)}{2} = \frac{d(d+3)}{2}.$$

(2). Lower bound for expected loss.

The infimum of the test loss over all possible algorithms $\mathcal{A}$ is

$$\inf_{\mathcal{A}} \mathop{\mathbb{E}}_{M \sim N(0,I_{d^2})} \left[ \mathop{\mathbb{E}}_{(\mathbf{x}_i, y_i) \sim P_X \diamond h_M} \left[ \ell_P(\mathcal{A}(\{\mathbf{x}_i, y_i\}_{i=1}^n)) \right] \right]$$

$$= \inf_{\mathcal{A}} \mathbb{E}_{M \sim N(0, I_{d^2})} \left[ \mathbb{E}_{(\mathbf{x}_i, y_i) \sim P_X \diamond h_M} \left[ \mathbb{E}_{\mathbf{x}, y \sim P_X \circ h_M} \left[ ([\mathcal{A}(\{\mathbf{x}_i, y_i\}_{i=1}^n)](\mathbf{x}) - y)^2 \right] \right] \right]$$

$$= \inf_{\mathcal{A}} \mathbb{E}_{M \sim N(0, I_{d^2})} \left[ \mathbb{E}_{\mathbf{x}_i \sim P_X} \left[ \mathbb{E}_{\mathbf{x} \sim P_X} \left[ ([\mathcal{A}(\{\mathbf{x}_i, h_M(\mathbf{x}_i)\}_{i=1}^n)](\mathbf{x}) - h_M(\mathbf{x}))^2 \right] \right] \right]$$

$$\geq \mathbb{E}_{\substack{\mathbf{x}_i, \mathbf{x} \sim P_X \\ M \sim N(0, I_{d^2})}} \left[ \operatorname{Var}_{\mathbf{x}, \mathbf{x}_i, M} \left[ h_M(\mathbf{x}) \mid \{\mathbf{x}_i, h_M(\mathbf{x}_i)\}_{i=1}^n, \mathbf{x} \right] \right]$$

$$= \mathbb{E}_{\substack{\mathbf{x}_i, \mathbf{x} \sim P_X \\ M \sim N(0, I_{d^2})}} \left[ \operatorname{Var}_{M} \left[ h_M(\mathbf{x}) \mid \{h_M(\mathbf{x}_i)\}_{i=1}^n \right] \right],$$

where the inequality is achieved when $[\mathcal{A}(\{\mathbf{x}_i, y_i\}_{i=1}^n)](\mathbf{x}) = \mathbb{E}_M [h_M(\mathbf{x}) \mid \{\mathbf{x}_i, y_i\}_{i=1}^n]$.

Thus it suffices to lower bound $\operatorname{Var}_M [h_M(\mathbf{x}) \mid \{h_M(\mathbf{x}_i)\}_{i=1}^n]$, for fixed $\{\mathbf{x}_i\}_{i=1}^n$ and $\mathbf{x}$. For convenience we define $\mathbb{S}^d = \{A \in \mathbb{R}^{d \times d} \mid A = A^\top\}$ be the linear space of all $d \times d$ symmetric matrices, where the inner product $\langle A, B \rangle := \operatorname{tr}[A^\top B]$ and $\Pi_n : \mathbb{R}^{d \times d} \to \mathbb{R}^{d \times d}$ as the projection operator for the orthogonal complement of the n-dimensional space spanned by $\mathbf{x}_i \mathbf{x}_i^\top$ in $\mathbb{S}^d$. By definition, we can expand

$$\mathbf{x} \mathbf{x}^\top = \sum_{i=1}^n \alpha_i \mathbf{x}_i \mathbf{x}_i^\top + \Pi_n(\mathbf{x} \mathbf{x}^\top).$$

Thus even conditioned on $\{\mathbf{x}_i, y_i\}_{i=1}^n$ and $\mathbf{x}$,

$$h_M(\mathbf{x}) = \operatorname{tr}[\mathbf{x} \mathbf{x}^\top] = \sum_{i=1}^n \alpha_i \operatorname{tr}[\mathbf{x}_i \mathbf{x}_i^\top M] + \operatorname{tr}[\Pi_n(\mathbf{x} \mathbf{x}^\top) M],$$

still follows a gaussian distribution, $N(0, \|\Pi_n(\mathbf{x} \mathbf{x}^\top)\|_F^2)$.

Note we can always find symmetric matrices $E_i$ with $\|E_i\|_F = 1$ and $\operatorname{tr}[E_i^\top E_j] = 0$ such that $\Pi_n(A) = \sum_{i=1}^k E_i \operatorname{tr}[E_i^\top A]$, where the rank of $\Pi_n$, is at least $\frac{d(d+1)}{2} - n$. Thus we have

$$\mathbb{E}_{\mathbf{x}} \left[ \|\Pi_n(\mathbf{x} \mathbf{x}^\top)\|_F^2 \right]$$

$$= \mathbb{E}_{\mathbf{x}} \left[ \left\| \sum_{i=1}^k E_i \operatorname{tr}[E_i^\top \mathbf{x} \mathbf{x}^\top] \right\|_F^2 \right]$$

$$= \sum_{i=1}^k \mathbb{E}_{\mathbf{x}} \left[ \|E_i \operatorname{tr}[E_i^\top \mathbf{x} \mathbf{x}^\top]\|_F^2 \right]$$

$$= \sum_{i=1}^k \mathbb{E}_{\mathbf{x}} \left[ (\mathbf{x}^\top E_i^\top \mathbf{x})^2 \right] \ (by \ Lemma \ D.8)$$

$$\geq \sum_{i=1}^k \|E_i\|_2^F \geq k$$

$$\geq \frac{d(d+1)}{2} - n$$

Thus the infimum of the expected test loss is

$$\inf_{\mathcal{A}} \mathbb{E}_{M \sim N(0, I_{d^2})} \left[ \mathbb{E}_{(\mathbf{x}_i, y_i) \sim P_X \diamond h_M} \left[ \ell_P(\mathcal{A}(\{\mathbf{x}_i, y_i\}_{i=1}^n)) \right] \right]$$

$$\geq \underset{\substack{\mathbf{x}_i,\mathbf{x}\sim P_X \\ M\sim N(0,I_{d^2})}}{\mathbb{E}} \left[ \underset{M}{\mathrm{Var}}\left[ h_M(\mathbf{x}) \mid \{h_M(\mathbf{x}_i)\}_{i=1}^n \right] \right].$$

$$= \underset{\substack{\mathbf{x}_i\sim P_X \\ M\sim N(0,I_{d^2})}}{\mathbb{E}} \left[ \underset{\mathbf{x}}{\mathbb{E}}\left[ \left\| \Pi_n(\mathbf{x}\mathbf{x}^\top) \right\|_F^2 \right] \right].$$

$$\geq \frac{d(d+1)}{2} - n.$$

**Upper bound:** We use the same CNN construction as in the proof of Theorem 5.3, i.e., the function class is $\mathcal{F}_{\mathsf{CNN}} = \left\{ \sum_{i=1}^d a_i w_1^2 x_i^2 + b | a_i, w_1, b \in \mathbb{R} \right\} = \left\{ \sum_{i=1}^d a_i x_i^2 + b | a_i, b \in \mathbb{R} \right\}$. Thus given $d+1$ samples, w.p. 1, $(x_1^2, x_2^2, \ldots, x_d^2, 1)$ will be linear independent, which means $\mathsf{ERM}_{\mathsf{CNN}}$ could recover the ground truth and thus have 0 loss.

$\square$

## D.7 Proof of Theorem 5.5

**Theorem 5.5.** *Let $\mathbf{t}_i = \mathbf{e}_i + \mathbf{e}_{i+1}$ and $\mathbf{s}_i = \mathbf{e}_i + \mathbf{e}_{i+2}$[4] and $P$ be the uniform distribution on $\{(\mathbf{s}_i, 1)\}_{i=1}^n \cup \{(\mathbf{t}_i, -1)\}_{i=1}^n$, which is the classification problem for local textures in a 1-dimensional image with $d$ pixels. Then for any permutation equivariant algorithm $\mathcal{A}$, $\mathcal{N}(\mathcal{A}, \mathcal{P}, \frac{1}{8}, \frac{1}{8}) \geq \mathcal{N}^*(\mathcal{A}, \mathcal{P}, \frac{1}{4}) \geq \frac{d}{10}$. Meanwhile, $\mathcal{N}(\mathsf{ERM}_{CNN}, \mathcal{P}, 0, \delta) \leq \log_2 \frac{1}{\delta} + 2$, where $\mathsf{ERM}_{CNN}$ stands for $\mathsf{ERM}_{CNN}$ for function class of 2-layer ConvNets.*

*Proof of Theorem 5.5.* **Lower Bound:** We further define permutation $g_i$ as $g_i(\mathbf{x}) = \mathbf{x} - (\mathbf{e}_{i+1} - \mathbf{e}_{i+2})^\top(\mathbf{e}_{i+1} - \mathbf{e}_{i+2})\mathbf{x}$ for $i \in [d]$. Clearly, $g_i(\mathbf{t}_i) = \mathbf{s}_i, g_i(\mathbf{s}_i) = \mathbf{t}_i$. For $i, j \in \{1, 2, \ldots, d\}$, we define $d(i, j) = \min\{(i - j) \bmod d, (j - i) \bmod d\}$. It can be verified that if $d(i, j) \geq 3$, then $g_i(\mathbf{s}_j) = \mathbf{s}_j, g_i(\mathbf{t}_j) = \mathbf{t}_j$. For $\mathbf{x} = \mathbf{s}_i$ or $\mathbf{t}_i, \mathbf{x}' = \mathbf{s}_j$ or $\mathbf{t}_j$, we define $d(\mathbf{x}, \mathbf{x}') = d(i, j)$.

Given $\mathbf{X}_n, \mathbf{y}_n$, we define $B := \{d(\mathbf{x}, \mathbf{x}_k) \geq 3, \forall k \in [n]\}$ and we have $\mathbb{P}[B] = \mathbb{P}_{\mathbf{x}}[d(\mathbf{x}, \mathbf{x}_k) \geq 3, \forall k \in [n]] \geq \frac{d - \frac{d}{10}*5}{d} = \frac{1}{2}$. Therefore, we have

$$\mathrm{err}_P(\mathcal{A}(\mathbf{X}_n, \mathbf{y}_n)) = \underset{\mathbf{x},y,\mathcal{A}}{\mathbb{P}}[\mathcal{A}(\mathbf{X}_n, \mathbf{y}_n)(\mathbf{x}) \neq y] \geq \underset{\mathbf{x},y,\mathcal{A}}{\mathbb{P}}[\mathcal{A}(\mathbf{X}_n, \mathbf{y}_n)(\mathbf{x}) \neq y \mid B]\,\mathbb{P}[B]$$

$$\geq \frac{1}{2}\underset{\mathbf{x},y,\mathcal{A}}{\mathbb{P}}[\mathcal{A}(\mathbf{X}_n, \mathbf{y}_n)(\mathbf{x}) \neq y \mid B]$$

$$= \frac{1}{4}\underset{i,\mathcal{A}}{\mathbb{P}}[\mathcal{A}(\mathbf{X}_n, \mathbf{y}_n)(\mathbf{s}_i) \neq 1 \mid B] + \frac{1}{4}\underset{i,\mathcal{A}}{\mathbb{P}}[\mathcal{A}(\mathbf{X}_n, \mathbf{y}_n)(\mathbf{t}_i) \neq -1 \mid B]$$

$$\overset{(3.2)}{=} \frac{1}{4}\underset{i,\mathcal{A}}{\mathbb{P}}[\mathcal{A}(g_i(\mathbf{X}_n), \mathbf{y}_n)(g_i(\mathbf{s}_i)) \neq 1 \mid B] + \frac{1}{4}\underset{i,\mathcal{A}}{\mathbb{P}}[\mathcal{A}(\mathbf{X}_n, \mathbf{y}_n)(\mathbf{t}_i) \neq -1 \mid B]$$

$$= \frac{1}{4}\underset{i,\mathcal{A}}{\mathbb{P}}[\mathcal{A}(\mathbf{X}_n, \mathbf{y}_n)(\mathbf{t}_i) \neq 1 \mid B] + \frac{1}{4}\underset{i,\mathcal{A}}{\mathbb{P}}[\mathcal{A}(\mathbf{X}_n, \mathbf{y}_n)(\mathbf{t}_i) \neq -1 \mid B] = \frac{1}{4}.$$

Thus for any permutation equivariant algorithm $\mathcal{A}$, $\mathcal{N}^*(\mathcal{A}, \{P\}, \frac{1}{4}) \geq \frac{d}{10}$.

**Upper Bound:** Take CNN as defined in Section 3.1 with $d' = d, r = 1, k = 2, \sigma : \mathbb{R}^d \to \mathbb{R}$, $\sigma(\mathbf{x}) = \sum_{i=1}^d x_i^2$, we have $\mathcal{F}_{\mathsf{CNN}} = \left\{ \mathrm{sign}\left[ a_1 \sum_{i=1}^d (w_1 x_i + w_2 x_{i-1})^2 + b | a_1, w_1, w_2, b \in \mathbb{R} \right] \right\}$.

Note that $\forall h \in \mathcal{F}_{\mathsf{CNN}}, \forall 1 \leq i \leq d, h(\mathbf{s}_i) = a_1(2w_1^2 + 2w_2^2) + b, h(\mathbf{t}_i) = a_1(w_1^2 + w_2^2 + (w_1 + w_2)^2) + b$, thus the probability of $\mathsf{ERM}_{\mathcal{F}_{\mathsf{CNN}}}$ not achieving 0 error is at most the probability that all data in the training dataset are $\mathbf{t}_i$ or $\mathbf{s}_i$: (note the training error of $\mathsf{ERM}_{\mathcal{F}_{\mathsf{CNN}}}$ is 0)

$$\mathbb{P}\left[\mathbf{x}_i \in \{\mathbf{s}_j\}_{j=1}^d, \forall i \in [n]\right] + \mathbb{P}\left[\mathbf{x}_i \in \{\mathbf{t}_j\}_{j=1}^d, \forall i \in [n]\right] = 2^{-n} \times 2 = 2^{-n+1}.$$

---

[4]For vector $\mathbf{x} \in \mathbb{R}^d$, we define $x_i = x_{(i-1) \bmod d+1}$.

**Convergence guarantee for Gradient Descent:**    We initialize all the parameters by i.i.d. standard gaussian and train the second layer by gradient descent only, i.e. set the LR of $w_1, w_2$ as 0. (Note training the second layer only is still a permutation-equivariant algorithm for FC nets, thus it's a valid separation.)

For any convex non-increasing surrogate loss of 0-1 loss $l$ satisfying $l(0) \geq 1, \lim_{x \to \infty} l(x) = 0$ e.g. logistic loss, we define the loss of the weight $\mathbf{W}$ as

$$\mathcal{L}(\mathbf{W}) = \sum_{i=1}^{n} l(\mathcal{F}_{\mathsf{CNN}}[\mathbf{W}](\mathbf{x}_i)y_i)$$
$$= N_S \times l\left(a_1(2w_1^2 + 2w_2^2) + b\right) + N_t \times l\left(-a_1(w_1^2 + w_2^2 + (w_1 + w_2)^2) + b\right).$$

Note $w_1 w_2 \neq 0$ with probability 1, which means the data are separable even with fixed first layer, i.e. $\inf_{a_1,b} \mathcal{L}(\mathbf{W}) = 0$. Further note $\mathcal{L}(\mathbf{W})$ is convex in $a_1$ and $b$, which implies with sufficiently small step size, GD converges to 0 loss solution. By the definition of surrogate loss, $\mathcal{L}(\mathbf{W}) < 1$ implies for $\mathbf{x}_i, l(\mathbf{x}_i y_i) < 1$ and thus the training error is 0.  $\square$

