# OpenReview forum: "Why Are Convolutional Nets More Sample-Efficient than Fully-Connected Nets?"
_ICLR.cc/2021/Conference — ICLR 2021 Oral_

### Official Review · AnonReviewer1 · 2020-10-25
**A nice work on the sample efficiency of ConvNets**

**Rating:** 7
**Confidence:** 4

**Review:**

This paper proves that, for the learning problem where the input distribution is standard Gaussian, and the ground-truth label is given by the difference between the sum of squares of the first half coordinates and second half coordinates, any orthogonal-equivariant algorithm (e.g., a fully-connected network with SGD) needs \Omega(d^2) samples to achieve a constant test error with a constant probability, while there exists a ConvNet which only needs O(1) samples. Similar results on l_2 regression and adaptive training algorithms are also given.

I think this paper attacks the important problem of sample efficiency of ConvNets. The notion of equivariance between algorithms, including orthogonal and permutation equivariance, is clean and inspiring: it can help us understand and improve various algorithms and architectures, and may also be extended to other settings. Various proof ideas are also interesting, such as Theorem 5.1, the use of Benedek-Itai's lower bound, etc. The writing is clean in my opinion.

Here is one question that is interesting to me: Suppose we insert a Gaussian-initialized fully-connected layer before a ConvNet, i.e., let the fully-connected layer be the first layer, followed by the original ConvNet. Now I think SGD on this new architecture becomes orthogonal equivariant, even if we don't train the first fully-connected layer. Does generalization deteriorate in this setting?

On the weakness of the paper, here are my thoughts:
1. For the proofs to work, it seems that the feature distribution needs to be rotational invariant. Can we relax this condition? On the other hand, it is mentioned that the target function is still easier for ConvNets to learn on CIFAR inputs, which partly answers this question.
2. The ConvNet used to learn the target function (given on page 15) is special: it is a nearly-minimal function class that can represent the target function. What if we use ReLU activation, max-pooling, etc.?

Here are some minor comments:
1. In the last inequality of Definition 3.3, the middle N^* should be N.
2. In the definition of ConvNets, the subscript d'(r-1)+1:d'r should be d'(i-1)+1:d'i.

---

> ### Author Response · Authors · 2020-11-25
> **Response to Reviewer 1**
>
> We thank you for your appreciation for our results and proof ideas! We will respond to your comments one by one.
>
> 1. **Does inserting an FC layer between input and ConvNets deteriorate generalization?** Yes. Our lower bound applies in that case. We also verified this in experiments. See updated Figure 2. This is indeed an excellent example to test our theory. Thank you for this thoughtful suggestion!
>
> 2. **Does the distribution have to be rotationally invariant for $\Omega(d^2)$ lower bound?** For the $\Omega(d^2)$  bound based Itai-Benedek's lower bound, the answer is yes. *However, it's possible to derive $\Omega(d^2)$ bounds for distributions not rotational invariant via different proof techniques,* e.g. the $\Omega(d)$ lower bound for permutation equivariance is proved for distribution which is not permutation equivariant. The proof is not based on Itai-Benedek's lower bound but by direct coupling instances between different permutations. We leave it as a future work to get a more general $\Omega(d^2)$ lower bound allowing distributions which is not rotation invariant.
>
> 3. **The upper bound is only proved for a very special CNN.** For the particular ConvNet(given on page 15), relu and Max pooling would not work as it cannot express the ground truth. For an over-parametrized ConvNet, expressiveness is not an issue because one can just use relu + bias to approximate quadratic activation and use many layers' convolution + max pooling to approximate a single global average pooling layer, while it may require more samples for generalization. In the revision, we show  **for our constructed hard instance, even more complicated ConvNet (e.g. Resnet with relu activation) still outperforms FC nets experimentally.**
>
>     In general, we believe it is an interesting and important research problem to identify and rigorously prove generalization for a broader class of ConvNets with suitable algorithms, but just orthogonal to the focus of this work. In this work, we use the fact that the hard instance could be learned by a very simple ConvNet to highlight the hardness brought by algorithmic equivariance.

---

### Official Review · AnonReviewer4 · 2020-10-28
**Interesting observation on the difference in optimisation of convnet ad MLP**

**Rating:** 7
**Confidence:** 3

**Review:**

The paper presents an interesting analysis of MLP and convnets, where they show a gap between the number of required training examples to generalize well. They show that due to orthogonality invariance in MLP training, then more examples are required compare to convnet, where one example is needed. This approach, which relies on an older result, provides an intuition as to the success of resnet.

While the work is interesting I have one main concern:
Is the distribution analyzed related to real problems? I think making such a relationship is important as at the moment I don't see any connection between the models analyzed and the structure of real data.
The reason this question is important is that in a similar way to the analysis performed, one may find a data distribution that cannot be learned with convnet but can be applied with MLP. Then convnet will get very bad error, while MLP will be able to generalize. So, it is important to explain why the distribution used in the analysis is related to realistic data.

---

> ### Author Response · Authors · 2020-11-25
> **Response to Reviewer 4**
>
> We thank you for your effort in reviewing our manuscript. Below is our response to your main concern.
>
> **Relation between our theory to real data distribution is not clear:** There is no easy math characterization of real-life datasets. The only mathematical (but approximate) characterization we have for what makes a collection of pixels a picture of a dog is the function represented by a Conv net. From this perspective, our hard distribution is related to real-life distributions in the sense that the labeling function for both of them can be expressed by Conv nets with much fewer hyper-parameters than FC nets. A good example is our hard instance for permutation invariance. The data distribution is translation invariant, which is a common property for vision tasks. And it's for this reason ConvNets can learn it with fewer samples.

---

### Official Review · AnonReviewer3 · 2020-10-31
**Interesting theoretical investigation, some points are unclear**

**Rating:** 7
**Confidence:** 4

**Review:**

The paper studies simple distributional settings in which convolutional neural networks give a provable sample complexity advantage over fully connected networks. This perspective is a valuable complement to prior work in statistical learning theory that often focuses on distribution-free results, which make it harder to study interactions between the training distribution and learning algorithm.

Overall I find this research direction very interesting and the paper is a good contribution. My two main concerns are the following:

- It is unclear how the proposed theory relates to real data distributions. Concretely:
  * The authors conduct two experiments, one of them with real examples (CIFAR-10 images) and synthetic labels (Figure 1). Did the authors explore a range of architectures and optimization hyperparameters for the fully-connected networks in Figure 1? Performance of neural networks can vary a lot under hyperparameter changes, so the separation would be more convincing if the authors performed a search through a space of hyperparameters.
  * In addition, it would be interesting if the authors related their theory to properties of real datasets (without synthetic labels).
  * The theoretical results (upper bounds) study the case where only the last layer of a neural network is trained. Is this also the case in Figure 1?

- The presentation of the theoretical results could be improved. Concretely:
  * The first eight pages contain about one page of definitions. While it is certainly important to be technically precise, some of the definitions could be moved to the appendix so that there is more space for conveying the core ideas in the main text (e.g., the next point).
  * The upper bounds for gradient descent on CNNs are for the convex case where only the second layer is trained. It would be good to state this in the main text so that the reader understands the flavor of the results more easily.
  * Some parts of the proofs in the appendix are only sketched or omitted, e.g., Lemma C.4 or the convergence guarantee for gradient descent in Theorem 4.1


Given these limitations, I am currently hesitant about accepting the paper even though I find the overall research questions very interesting. Addressing the points above could substantially improve the paper.


Additional comments:

- Page 21: the step from the second to last line to the last line is unclear to me. We have conditioned on the event B, which presumably means d(x, x_i) >= 3  (the event B is not clearly defined?). Hence we should have tau_i(s_i) = s_i, but the last line replaces tau_i(s_i) with t_i?

- Definition 3.3 contains "n \in ?" - what does this symbol stand for?

- Page 5: typo "bewteen"

- Theorem 4.2: typo "samples from a fixed ,"

- The appendix contains many proofs without restating the theorems from the main text, which makes it hard to read the paper thoroughly. The thm-restate package is helpful for this, see https://tex.stackexchange.com/questions/51286/recalling-a-theorem

--------------------------------------------------------------------------------------------

Thank you for addressing my comments, I have updated my score accordingly.

---

> ### Author Response · Authors · 2020-11-25
> **Response to Reviewer 3**
>
> Thank you for your detailed reviews and constructive suggestions. We will respond to your comments one by one.
>
> 1. **Does the experimental result hold over a wide range of architectures of FC nets:** Yes. We show the lower bound for FC nets hold for various FC architectures. See details in Figure 2. We didn't find a significant difference in performance by using different learning rates so we didn't report them.
>
> 2. **Relation between our theory to real data distribution is not clear:** There is no easy math characterization of real-life datasets. The only mathematical (but approximate) characterization we have for what makes a collection of pixels a picture of a dog is the function represented by a Conv net. From this perspective, our hard distribution is related to real-life distributions in the sense that the labeling function for both of them can be expressed by Conv nets with much fewer hyper-parameters than FC nets. A good example is our hard instance for permutation invariance. The data distribution is translation invariant, which is a common property for vision tasks. And it's for this reason ConvNets can learn it with fewer samples.
>
> 3. **Is only the last layer of the CNN trained in Figure 1?** No. Both two layers are trained.
>
> 4. **Improvement suggestions for presentation of theoretical results**: We adopt your suggestions, including providing the original omitted/sketched proofs and stating in the main text that GD is only on the second layer.
>
> 	We also want to clarify that the upper bound is indeed for ERM, not for GD. The reason we mention GD for the second layer can implement ERM is just to make a rigorous separation,  i.e.,  all  FC  net trained by  GD  vs some  CNN  trained by GD. We are not claiming any technical contribution for these simple upper bounds.  Rather, we use them as a comparison to highlight the hardness brought by algorithmic equivariance.
>
> 5. **Confusion on Page 21**: We apologize for the typo in this proof --- the $\tau_i$ are indeed $g_i$ defined above. We fixed this in the revision. By definition, $g_i$ keeps $s_j,t_j$ unchanged, but transforms $t_i$ into $s_i$, $s_i$ into $t_i$ when $d(i,j)>3$; vice versa.
>
> We also fixed other typos you mentioned and used thm-restate package. We hope that you will reconsider your ratings to reflect the revisions and the responses.

---

### Official Review · AnonReviewer2 · 2020-10-31
**Review for Why Are Convolutional Nets More Sample-Efficient than Fully-Connected Nets?**

**Rating:** 8
**Confidence:** 4

**Review:**

This paper studies an interesting theoretical question: are there any natural tasks that provably separate fc-nets from convnets. The main contribution of this paper is an Omega(d^2) vs O(1) separation.

To prove the hardness result, the authors use (and generalize) the notion of orthogonal-equivariance introduced by Ng (2004). The current submission improves the hardness results of Ng (2004) in the following aspects:

1. Ng (2004) proved an Omega(d) vs O(1) separation, while this paper provides an Omega(d^2) vs O(1) separation. This is interesting not only from a theoretical perspective, but could also be relevant to practice. In practice, the dimensionality d is always moderately large. Moreover, the labeling function employed in the hard case is natural and could indeed capture practical scenarios.
2. The hardness result by Ng (2004) does not use a fixed hard distribution, while this paper shows that there exists a universal (and in fact, natural and simple) hard distribution that is hard for any orthogonal-equivariant algorithm. Personally I find such an improvement important: in order to demonstrate the intrinsic superiority of convnets over fc-nets, it is crucial to obtain distributions that are hard for all training algorithms.
3. The authors generalize the notion of orthogonal-equivariance and propose permutation-invariance, which allows them to prove hardness results for a wider class of algorithms. In particular, separation between fc-nets and convnets trained by Adam, which is a corollary of the hardness result in this paper, is not implied by previous results. Generalizing hardness results to a larger class of algorithms is definitely interesting for a broad class of audience in the deep learning community.

The lower bound is proved by using Benedek-Itai’s approach and carefully bounding certain covering numbers.

Overall, this paper presents a set of interesting results which rigorously explain why covnets could be more sample-efficient than fc-nets. This paper is generally well-written and provides lots of intuition on why the hardness results hold. On the other hand, there are a few typos that need to be fixed (see below). Given the great importance of the topic and results in this paper, I would recommend acceptance.


Minor Comments:

For random variable X and Y => random variables

for function class F,G => function classes. also missing space before G

semi-definite positive matrix => positive semi-definite matrix

ConvNets(CNN): missing space

( which may depend on S) : extra space

models below , FC-NN: extra space

But as noted in the introduction: remove but

namely, the standard Gaussian, => the standard Gaussian distribution

---

> ### Author Response · Authors · 2020-11-25
> **Response to Reviewer 2**
>
> We thank you for your appreciation! We've fixed all typos you mentioned.

---

### Author Response · Authors · 2020-11-25
**An Overview of Paper Update**

We thank all the reviewers sincerely for their constructive and positive feedback. We have incorporated the suggestions in our updated manuscript. Below we provide an overview of the revisions. Please also see our individual responses to each reviewer.

Major Updates:

1. We added a new figure (Figure 2) in appendix section E, where we compare the test accuracy of networks with a broader class of architectures, including Resnet, ReLu activation, and BatchNorm, on the same hard instance used in Figure 1. We found that ConvNets still always outperform FC nets experimentally.

2. In Figure 2, we also reported the performance of a hybrid architecture of FC and Conv nets --- we add an FC layer before the ConvNet with quadratic activation, as suggested by Reviewer 1. Not surprisingly, as predicted by our lower bounds, the performance of the hybrid net is as low as FC nets.

---

### Decision · Program_Chairs · 2021-01-07
**Final Decision**

**Decision:**

Accept (Oral)

**Comment:**

The paper analyzes the sample complexity of convolutional architectures, proving a gap between it and that of fully connected (fc) networks. The approach builds on certain invariances of fc nets. The reviewers appreciated the technical content and its contribution to understanding the relative advantages of different architecture, as well as the role of invariance.